# Emergent biaxiality in chiral hybrid liquid crystals

Jin-Sheng Wu [1], Marina Torres Lázaro [2], Haridas Mundoor[1], Henricus H. Wensink[2] & Ivan I. Smalyukh [1,3,4,5] ✉

Biaxial nematic liquid crystals are fascinating systems sometimes referred to as the Higgs boson of soft matter because of experimental observation challenges. Here we describe unexpected states of matter that feature biaxial orientational order of colloidal supercritical fluids and gases formed by sparse rodlike particles. Colloidal rods with perpendicular surface boundary conditions exhibit a strong biaxial symmetry breaking when doped into conventional chiral nematic fluids. Minimization of free energy prompts these particles to orient perpendicular to the local molecular director and the helical axis, thereby imparting biaxiality on the hybrid molecular-colloidal system. The ensuing phase diagram features colloidal gas and liquid and supercritical colloidal fluid states with long-range biaxial orientational symmetry, as supported by analytical and numerical modeling at all hierarchical levels of ordering. Unlike for nonchiral hybrid systems, dispersions in chiral nematic hosts display biaxial orientational order at vanishing colloid volume fractions, promising both technological and fundamental research utility.

Since the experimental discovery of chiral nematic liquid crystals (LCs)[1,2], LC mesophases featuring chirality and long-range orientational order have been the focus of many research studies. Chiral nematic LCs as model systems provide extensive insights into physics principles associated with experimentally less accessible systems like particle physics or cosmology[3–12], in addition to their technological applications in electro-optics and displays. On the other hand, biaxial nematic mesophases have been highly sought-after in soft matter systems since their first theoretical consideration in 1970[13]. However, even in a soft-matter system with strongly biaxial building blocks such as brick-shaped molecules, biaxiality was experimentally elusive and often hard to unambiguously demonstrate in equilibrium states[14,15]. Recent experimental studies of biaxial nematic order include observations in colloidal dispersion of highly anisotropic particles immersed in molecular LC hosts, so-called hybrid molecular-colloidal nematics[16–18]. The interplay between chirality and biaxiality in orientational order has been intensively studied[19–28]. However, for purely molecular or purely colloidal systems, the chirality-induced biaxiality of the molecular orientation distribution was predicted and experimentally found to be rather weak[19–26], scaling as $(qL_m)^2$ according to the prediction by Priest and Lubensky for single-component molecular LCs[19], where $q = 2\pi/p$, $p$ is the helical pitch of the chiral nematic (typically in microns range) and $L_m$ the molecular length (about 1nm). To date, there are no considerations on whether the biaxial symmetry of the orientational distribution of anisotropic colloidal particles could interplay with the chirality of the nematic host in hybrid molecular-colloidal LC systems, or whether emergent effects beyond the chirality-induced biaxiality at the molecular level could possibly arise. While combining nematic hosts with strongly anisotropic colloidal particles allowed for the formation of novel monoclinic, orthorhombic, and other unexpected mesophases in the ensuing hybrid molecular-colloidal LCs[16–18], no analogous research

[1]Department of Physics and Chemical Physics Program, University of Colorado, Boulder, CO, USA. [2]Laboratoire de Physique des Solides - UMR 8502, Université Paris-Saclay & CNRS, Orsay, France. [3]International Institute for Sustainability with Knotted Chiral Meta Matter, Hiroshima University, Higashihiroshima, Japan. [4]Department of Electrical, Computer, and Energy Engineering, Materials Science and Engineering Program and Soft Materials Research Center, University of Colorado, Boulder, CO, USA. [5]Renewable and Sustainable Energy Institute, National Renewable Energy Laboratory and University of Colorado, Boulder, CO, USA. ✉e-mail: ivan.smalyukh@colorado.edu

studies have been reported to date for the chiral counterparts of these soft matter systems.

In this work, we combine experiments, numerical simulations and theory to demonstrate emergent as well as strongly enhanced biaxial order for uniaxial colloidal particles dispersed in a chiral nematic molecular host. Anisotropic colloidal inclusions interact with the geometry of the chiral LC with three non-polar director fields (Fig. 1): molecular director field $\hat{\mathbf{n}} = -\hat{\mathbf{n}}$ representing the local average molecular alignment, the helical axis field $\hat{\chi} = -\hat{\chi}$ relative to which $\hat{\mathbf{n}}$ rotates, and $\hat{\tau} = \pm\hat{\mathbf{n}} \times \hat{\chi}$[29,30]. Unlike in nematic hybrid molecular-colloidal LCs[18], where biaxial order was only found to emerge beyond a critical colloidal particle concentration, we find that the orientational order of anisotropic colloidal inclusions in chiral nematic hosts is strongly biaxial even at very low colloid particle volume fractions (Fig. 2).

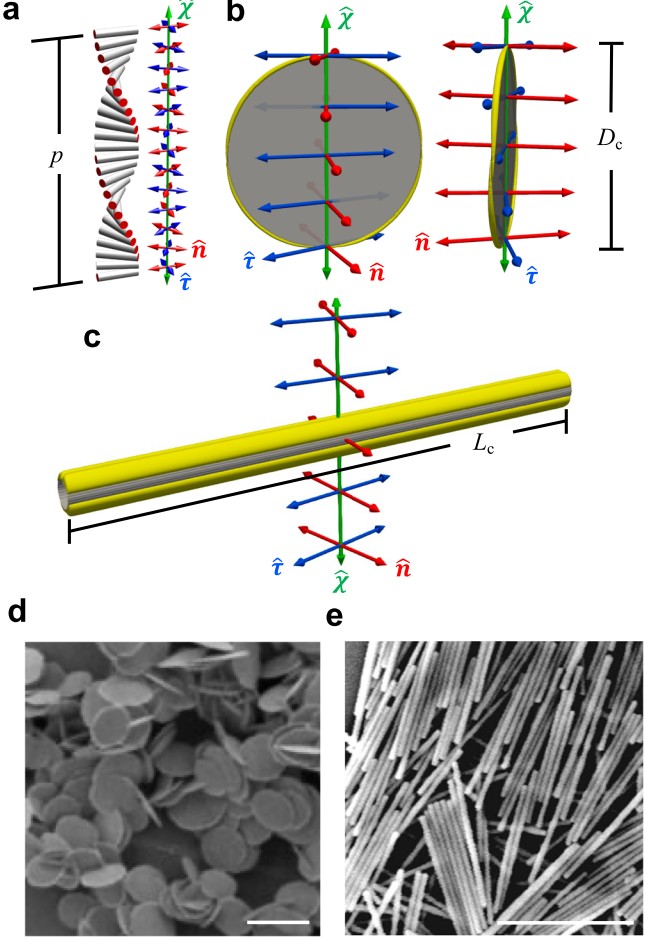

**Fig. 1 | Chiral hybrid liquid crystals. a** Helical structure of a chiral LC with helical pitch length $p$, with cylinders representing LC molecules and colored axes depicting the orthogonal cholesteric frame: LC molecular director $\hat{\mathbf{n}}$ (red), helical axis/director $\hat{\chi}$ (green), and the third axis/director $\hat{\tau}$ (blue) orthogonal to both. **b**, **c** Visualizations of a homeotropic ($\hat{\mathbf{n}}$ tending to orient perpendicular to the colloidal particle's surface) colloidal disk (**b**) and rod (**c**) immersed in a chiral LC at their equilibrium orientations. Colloidal particles are depicted in gray, and the yellow contours mark a director deviation of 0.67° (**b**) and 0.3° (**c**), respectively, of the numerically energy-relaxed LC structures from the ideal helical state indicated by the colored double arrows. For all simulations the anchoring at the colloid surface is homeotropic with strength $W_0 = 10^{-4} Jm^{-2}$ (see "Methods" section). Typical disk diameter $D_c = 1\,\mu m$ and rod length $L_c = 1.7\,\mu m$ are more than one order of magnitude smaller than the pitch $p = 30\,\mu m$ of the chiral host LC. **d**, **e** TEM of synthesized disks and SEM of rods, scale bars 2 $\mu m$.

The most striking example of such emergent biaxiality is found for rods that tend to orient perpendicular to the local molecular director, imparting hierarchical biaxiality at molecular, colloidal, and structural levels of the hybrid colloid-molecular system. At the colloidal sub-system level, a generic phase diagram spanning the colloid concentration, temperature, and chirality strength unexpectedly features colloidal gas and liquid states, both with an orthorhombic long-range orientational order interplaying with that of the chiral nematic LC host. The colloidal liquid-gas phase coexistence terminates in a gas-liquid type critical point located at a well-defined chiral strength of the molecular host (Fig. 2e, f).

While the quest for stable biaxial nematic LCs[13] stimulated decades of intense experimental and theoretical research[14–18], where this mesophase was sometimes referred to as the Higgs boson of soft matter because of challenges in proving its existence, the possibility of generating stable biaxial orientational order within a sparse colloidal gas or colloidal supercritical fluid dispersed in a molecular cholesteric host was never considered. In this paper, we unambiguously demonstrate such a scenario for dilute dispersions of colloidal rods immersed in a structured cholesteric LC with uninhibited 3D fluidity, with the long-range biaxial orientational ordering revealed experimentally (Fig. 2). Biaxial symmetry breaking arises locally, since the two directions perpendicular to the average direction of colloidal alignment are not equivalent, and globally on the hybrid scale because the molecules and colloidal rods point along mutually orthogonal directors, both being orthogonal to the helical axis. To benchmark our findings with a more conventional scenario, we also study colloidal disks that tend to co-align with their normals along the local molecular director[19–28]. We show that, in addition to the emergent effects described above, even the helical-axis-mediated interplay between chirality and biaxiality in hybrid molecular-colloidal LCs exceeds levels that are found in purely molecular or colloidal systems. Finally, we discuss how our findings may allow for expanding the use of chiral molecular-colloidal LCs as model systems in studies of defects and topological solitons hosted by states of matter with high-dimensional order parameter spaces.

## Results

### Supercritical colloidal fluids with emergent biaxiality

Although colloidal analogs of states of matter like crystals, liquids, and gases are commonly studied, our findings here reveal orientationally ordered states with biaxiality exhibited by sparse dispersions of particles that have no analogs in molecular or atomic systems. Thin cylindrical colloidal rods with homeotropic (perpendicular) boundary conditions immersed in a chiral molecular LC correlate their orientations and form a helicoidal structure of rotating rods locally aligned orthogonal to $\hat{\mathbf{n}}$, revealing robust long-ranged colloidal orientational order even for well-isolated sparse rods at low molecular chirality (Fig. 2a, b). The biaxiality of these hybrid LC systems with helical structures of both molecules and homeotropic rods arises for various rod concentrations as the rods locally tend to orient, on average, along the $\hat{\tau}$ axis of the chiral nematic LC host for all observed rod densities (Fig. 2a–c). The mutually orthogonal fields, molecular and colloidal directors along with the helical axis, thus characterize the orthorhombic biaxiality for our composite system with a long twisting period $p$. By defining an angle correlation function as $\langle \frac{3}{2}\cos^2(\eta(r_0) - \eta(r_0 + r)) - \frac{1}{2} \rangle$ with $\langle . . . \rangle$ denoting an ensemble average[31], we also observe strong $\eta$ correlations of rod orientations persisting at large distances (quantitatively measured for up to 100 $\mu m$) in our experiments with an average rod length being 1.7 $\mu m$, illustrating orientationally ordered colloidal rods with angular distribution strongly coupled to the cholesteric frame (Fig. 2d).

In a nonchiral nematic LC host with $q = 0$, in contrast, the same colloidal rods are free to rotate around $\hat{\mathbf{n}}$, so that all azimuthal orientation angles $\eta$ are energetically identical, indicating a uniaxial orientational distribution. We find that $\eta$ for such cylindrical rods immersed

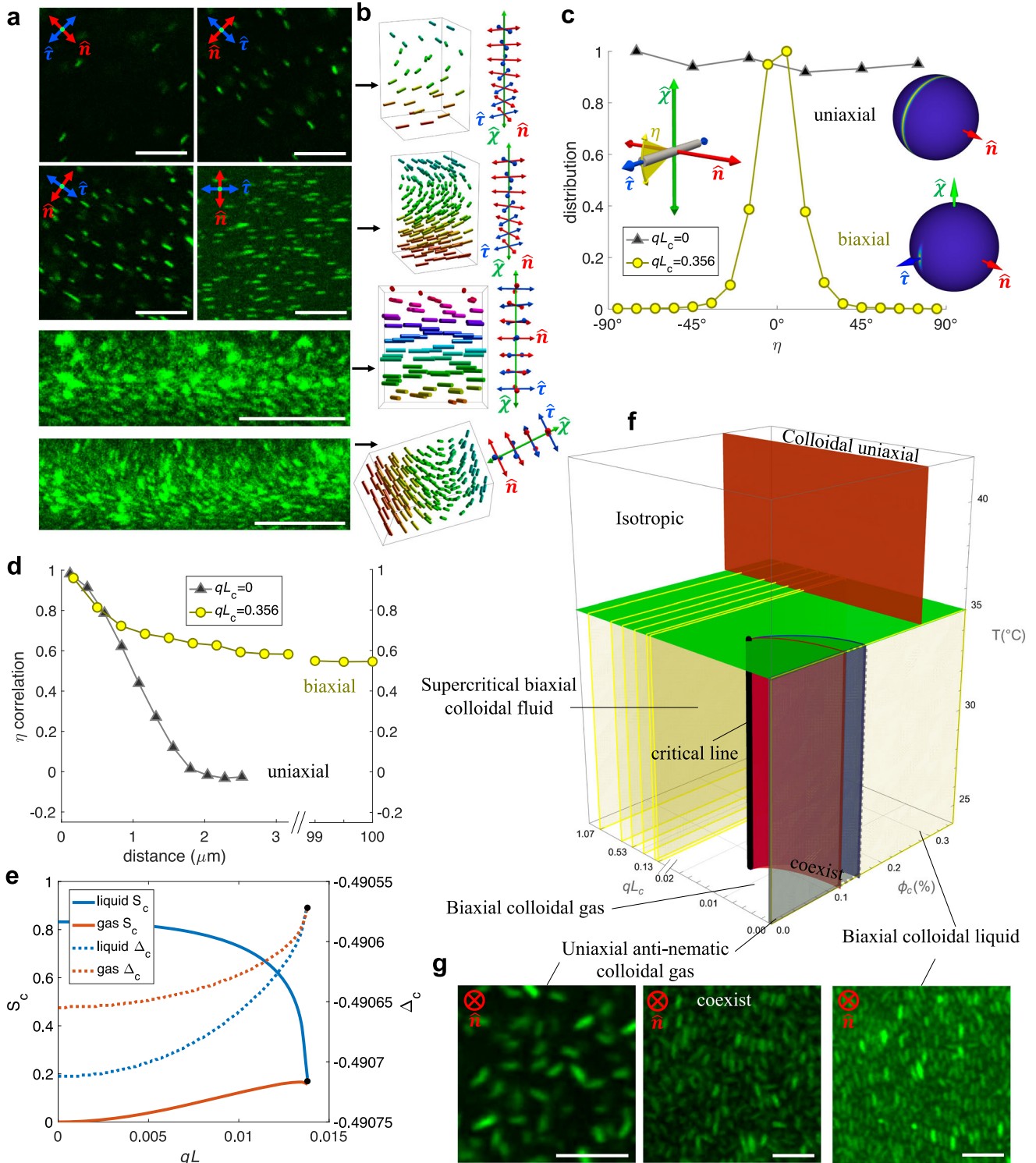

**Fig. 2 | Supercritical biaxial colloidal fluid. a** Cross-sectional micrographs of chiral hybrid LC samples with colloidal rods rendered green. The directors ($\hat{n}, \hat{\chi}, \hat{\tau}$) are marked on the top four images. Images of helical arrangements of rods obtained for oblique viewpoints are shown at the bottom. Scale bars are 10 μm, cholesteric pitch $p = 30$ μm and rod length $L_c = 1.7$ μm. **b** Schematic presentation of the supercritical biaxial colloidal fluids in (**a**) with rods colored by their orientations. The right-side insets show corresponding co-rotating cholesteric frames. **c** Distributions of colloidal angles $\eta$, defined as the angle in $\hat{\tau}\cdot\hat{\chi}$ plane (left inset), measured for low-concentration rod dispersions in nematic LC (black triangle) or cholesteric LC (yellow circle), with $q = 2\pi/p$ being the wavenumber associated with molecular chirality. The corresponding orientation distributions of rods in the cholesteric frame are shown as insets on the right side (see Fig. 8 for details). Colloidal particle concentration $\phi_c \ll 0.1\%$. **d** Corresponding rod angle $\eta$ correlation function (1 for

perfectly aligned and 0 for uncorrelated, see "Results" section) measured for the same pair of samples as in **c**, showing robust long-range correlations of colloidal orientations in cholesteric hosts at $qL_c = 0.356$. **e, f** Theoretical prediction of the colloidal $S_c$ and $\Delta_c$ order parameters, quantifying the orientational order with respect to $\hat{n}$ for coexisting phases of rod dispersions with perpendicular boundary conditions within the chiral LC host. The plots in (**e**) are order parameters quantified along the phase boundaries colored similarly in (**f**), up to the critical molecular chirality ($qL_c \approx 0.014$, marked black) beyond which the biaxial colloidal liquid and gas phases are no longer distinguishable. Temperature $T$, colloidal volume fraction $\phi_c$, and scaled molecular chirality $qL_c = 2\pi L_c/p$. Our experiments have been performed for parameters corresponding to the yellow planes with $qL_c = 0–1.07$ and $\phi_c = 0–0.35\%$. **g** Images of LC samples at $qL_c = 0$ showing an anti-nematic colloidal gas, gas-liquid coexistence, and a biaxial colloidal liquid phase, respectively. Scale bars are 5 μm.

in a non-chiral nematic LC (differently from its chiral counterpart) become random once they are separated by more than the rod length 1.7 μm, leading to uncorrelated rod orientations when the colloidal particle concentration is low and direct rod-rod interactions can be ignored (Fig. 2c, d).

In order to gain further insight into the physical origins of our experimental results, we developed a mean-field theoretical model describing the molecular-colloidal hybrid chiral LC system. The model predicts a subtle colloidal-concentration-dependent phase behavior driven by chirality as well as the corresponding orientational order parameters (Fig. 2e, f). Further details on the order parameters are given in the Results section and in the Supplementary Information. In the limiting cases, the predicted behaviors are consistent with a concentration-induced uniaxial-biaxial ordering transition within nonchiral nematic LC hosts ($qL_c = 0$)[18] and with the Priest-Lubensky model describing the emergence of weak biaxiality[19] in pure cholesterics ($\phi_c = 0$) without colloidal inclusions (Fig. 2f, g). Contrasting with the previously studied biaxiality of pure cholesterics and nonchiral molecular-colloidal LCs, we find that a chiral molecular-colloidal LC features a supercritical biaxial fluid regime (Fig. 2e,f) consistent with our experimental observation of strongly biaxial orientational correlations exhibited by the dispersed colloids (Fig. 2a–d).

## Biaxial colloidal gas–liquid coexistence

In our phase diagram (Fig. 2f), we introduce the notion of a colloidal gas to describe colloidal dispersions with large interparticle separation distances and negligible direct inter-colloidal interactions, much like colloidal gases are introduced in conventional colloidal systems of sparse dispersed particles. Similarly, hybrid LCs with higher $\phi_c$ showing aligned anisotropic colloidal objects are termed colloidal liquid, which is the orthorhombic biaxial nematic studied earlier[18], whereas supercritical biaxial colloidal fluid state is called so in analogy to conventional supercritical fluids that for chiral nematic hosts can exist at different varying colloidal densities without a phase boundary between the dense liquid-like and sparse gas-like colloidal states. The theoretical phase diagram was inspired and calibrated (Supplementary) by correlating experimental findings like the ones seen in images in Fig. 2a and g with the model predictions (Supplementary), and then tested further by performing additional experiments through varying all the parameters. Similar to our previous studies of non-chiral hybrid molecular-colloidal nematic systems[18], experimental findings of phase boundaries are consistent with the model at $qL_c = 0$ and for $qL_c = 0.13 − 1.07$ that we could probe experimentally, for concentrations and temperatures displayed in the diagram (Fig. 2f). At the same time, the model reveals interesting behavior in the vicinity of the critical line in the range of $qL_c$ that so far could not be probed experimentally. While the described here analogy with conventional gas-liquid coexistence relies on the fact that there is coexistence between two phases with an identical global point-group symmetry (orthorhombic $D_{2h}$ in our case) differing only in their number density, there are some important differences that we wish to point out.

The first difference is that a conventional gas-liquid transition usually entails long-ranged cohesive forces between particles which, in the classical van der Waals picture, give rise to a coexistence between a macroscopic gas (or vapor) and liquid phase below a critical temperature. In our systems, however, phase coexistence is primarily driven by entropic effects imparted by short-range repulsive anisotropic interactions between the colloids. Attractive centre-of-mass forces between the rod colloids, for instance those transmitted by tiny director distortions surrounding the rod surface, are too weak and short-ranged to have any impact (see Fig. 3). The second difference is a subtle coupling between particle density and alignment of the constituents' principal directions which is usually considered irrelevant for weakly anisotropic atoms, molecules or colloids with long-range attractive forces undergoing gas-liquid phase separation. The basic

mechanism underpinning the phase transition in our systems has been proposed by Onsager in his classic paper[32], namely a trade-off between orientational and excluded-volume entropy. Upon increasing the colloid concentration rods give up some of their orientational freedom by aligning more strongly in order to afford more free volume and reduce their mutual excluded-volume repulsion. As a consequence, the coexisting gas and liquid phases do not possess the same degree of orientational order (Fig. 2e). In the corresponding experimental images the coexistence region features domains of ordered rods next to domains with disordered ones (Fig. 2g).

In a hybrid molecular-colloidal system, the forces acting on the colloids are compounded by the presence of surface-anchoring torques imparted by the molecular host which restrict their rotational freedom. The extent of angular restriction experienced by the colloids scales with the chiral strength of the host such that increasing $qL_c$ lowers the threshold concentration at which the phase transition happens. At a critical chiral strength ($qL_c = 0.014$) the rods become energetically locked and the orientational entropy of the rods is too severely restricted to leave room for any Onsager-type entropy trade-off and the phase transition ceases to exist. In the supercritical regime $qL_c > 0.014$ there is only a single biaxial colloidal fluid and no phase coexistence is possible.

A rough estimate of the critical chiral strength can be obtained by equating the typical energy scale imparted by the surface-anchoring energy landscape (see Methods) to the thermal energy associated with the loss of orientational entropy between the gas and liquid phases. Assuming perpendicular surface anchoring to be sufficiently strong and ignoring prefactors of $\mathcal{O}(1)$, we find that the critical pitch of the LC host depends on the colloidal length $L_c$ and elastic anisotropy of the host as follows:

$$q_{crit} \sim \sqrt{\frac{k_B T}{(K_{33} − K_{11})L_c^3}} \tag{1}$$

with $k_B T$ the thermal energy in terms of temperature $T$ and Boltzmann's constant $k_B$ and $K_{11}$ and $K_{33}$ respectively denoting the splay and bend elastic moduli of the molecular LC. Taking $K_{33} − K_{11} = 4$ pN and $L_c = 1.7$ μm we find $qL_c \approx 0.02$ in good agreement with the value found numerically. Note that the critical pitch scales as $L_c^{-3/2}$ which suggests that the biphasic region in Fig. 2f would be much more extended for short colloidal rods. Given the relatively narrow temperature interval probed in experiment the critical line only weakly varies with temperature (Fig. 2f). Experimentally, the existence of the critical line is directly implied by the presence of the gas-liquid coexistence region at $qL_c = 0$ whereas at elevated chirality $qL_c > 0.13$ the hybrid system no longer features a distinct colloidal gas and liquid or co-existence region but only a uniform fluid state, resembling the transformation of a gas-liquid transition into a supercritical fluid state in conventional liquids upon increasing temperature (Fig. 2e, f). Probing details of the phase diagram in the vicinity of the critical line will require much smaller colloidal rods. To achieve the critical transition point at $qL_c = 0.014$ with our synthesized rods with $L_c = 1.7$ μm, one would need a chiral LC with a helical pitch in the near millimeter range in cells with even larger thickness but making such samples with monodomain alignment proves impractical due to an abundance of defects and limited effectiveness of surface alignment layers. To probe the phase transition behavior around the critical line one could instead use colloidal rods shorter than 100nm, the orientations of which are, however, challenging to resolve with videomicroscopy due to the optical resolution constraints. While such studies can be potentially done, for example, with plasmonic nanorods[33] or carbon nanotubes[34] by exploiting polarization-dependent surface plasmon resonance or luminescence spectra, respectively, suitable chiral nematic colloidal dispersions of such nanoparticles will still need to be developed.

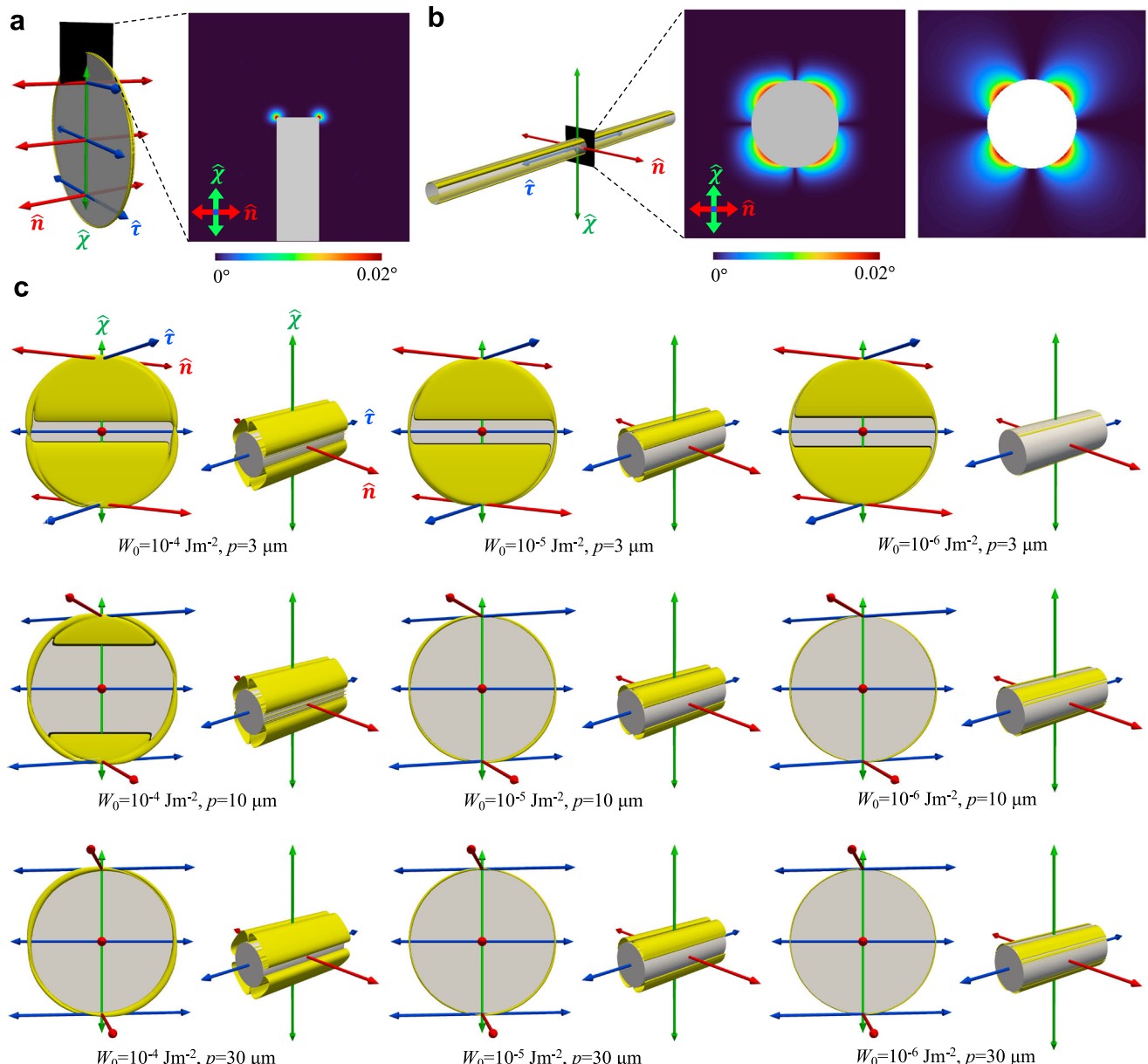

**Fig. 3 | Director distortions from numerical simulation. a**, **b** Computer simulations of a thin colloidal disk (**a**) and a slender rod (**b**) immersed in a LC with weak chirality. Yellow contour surfaces mark the region where LC director deviations for $\hat{n}$ (red axis) are 0.01° from its ideal helical state with no colloids present. The sectional areas perpendicular to $\hat{\tau}$ (blue axis) are numerical simulation results (left-side cross-section) colored by the deviation angle as shown in the color scale. The image on the right shows the corresponding analytical prediction (see Results). Homeotropic anchoring $W_0 = 10^{-6}$ Jm$^{-2}$ and a helical pitch $p = 30$ μm are used for all

calculations. **c** Simulations of colloidal disks and rods in energy-minimizing orientations within chiral LCs using various anchoring strengths $W_0$ and pitch lengths $p$, with values labeled for each simulation. Yellow contours enclose regions with director distortions larger than or equal to 0.1°, showing different levels of weak biaxiality. Rods in (**c**) are cropped for clarity. Axes defining the cholesteric frame are colored as in Fig. 1. Disk width $D_c = 1$ μm and rod length $L_c = 1.7$ μm for all simulations.

We finally wish to draw an analogy with disorder-order transitions explored in rod-shaped particles under the influence of an external electro-magnetic field or other external stimulus that acts on the rod orientations alone[35,36]. The overall structure of the phase diagram is similar to ours with a phase transition from a dilute para-nematic to dense nematic moving toward smaller concentrations as the external field grows stronger and a biphasic-gap narrowing into a critical point at a specific field strength[35,36]. A crucial difference with the external field case, besides the fact that the (para-)nematic phases in that case are uniaxial rather than biaxial, is that in our systems the orientational bias is transmitted entirely by the internal chiral symmetry of the molecular host which introduces an energy landscape favoring rods to

orient along the $\hat{\tau}$ axis rather than the helical $\hat{\chi}$ axis (Fig. 2c). This gives rise to strong biaxial symmetry breaking in the colloidal orientational distributions. The following section details how the degree of host LC chirality, quantified by $q$, imposes orientational constraints onto the colloidal rods which leads to the emergence of a supercritical biaxial colloidal fluid found in our experiments.

## Origins of the biaxial colloidal supercritical fluid

In order to explore the origins of the biaxial symmetry in our chiral hybrid LC system, we investigate the symmetry-breaking behavior, induced by the twisted alignment of chiral molecules, as well as of the nematic colloidal geometry at the single particle level (or colloidal gas

phase) by visualizing the LC distortion field around the particle (Fig. 3). Here, the LC distortion is the local realignment of $\hat{n}$ of the LC host due to the surface anchoring effect on colloidal-molecular interfaces and can be revealed by numerical simulation of the host medium ("Methods" section). When the cylindrical colloids are dispersed into a chiral nematic host, the uniaxial symmetry is broken in view of the boundary condition at particle-molecule interfaces and the far-field helical configuration of the LC molecules. Even when the homeotropic boundary conditions at the colloidal surfaces are rather weak with $W_0 = 10^{-6}$ J/m² and deviation angles relative to the easy axis are small, $\ll 1°$ (Fig. 3a,b), for example, it is evident that the rotational symmetry of the surface-defect-dressed cylindrical colloids becomes discrete (2-fold rotation) once the colloids are immersed in a cholesteric host (Fig. 3c). Clearly, stronger surface anchoring forces and higher chirality (shorter pitch) lead to significantly stronger molecular LC distortion as well as the ensuing emergent biaxiality as shown by the computer-simulated distortion in nematic director. Also, the single-particle symmetry-breaking is observed even when the helical pitch $p$ is much larger than the particle dimension, with the particle sizes around 1–2 μm (Fig. 3c). Importantly, both rods and disks induce corona of molecular alignment perturbations of low symmetry, lower than the uniaxial symmetry of particles themselves (Fig. 3). This demonstrates that the shape biaxiality of the dressed colloidal particle imparted by the molecular chirality of the host is unavoidably developed at all strengths of surface anchoring and values of molecular chirality (Fig. 3c). We shall see below that the low-symmetry deformations of molecular alignment around individual particles generate a chirality-dependent orientational energy landscape, leading to the supercritical behaviors of the colloidal phases.

## Computational analysis of particle orientations

Since the surface-induced molecular alignment structures around a particle and associated energies are not uniaxial, the orientational fluctuations of such particles experiencing those symmetry-breaking energy landscapes naturally pick up the biaxial symmetry. To analyze the orientational distribution of the cylindrical particles, we accordingly perform several sets of numerical simulations at various colloidal orientations and resolve the corresponding free energies (Methods). By employing mean-field numerical simulations of the LC host, we are able to validate the local biaxial symmetry of the orientational probability distributions of the individual colloids, which arises from the energetic inequivalence of colloidal orientations along $\hat{\chi}$ and $\hat{\tau}$ in the molecular LC host.

For example, a homeotropic rod feels a strong energy penalty when rotated to point along $\hat{n}$ and reaches a state of minimal surface anchoring energy when the long axis points along the $\hat{\tau}$-direction such that the LC director at the rod surface naturally complies with the homeotropic surface anchoring conditions (Fig. 4)[18,37]. The symmetry-breaking of $\hat{\chi}$ and $\hat{\tau}$, evident from the difference between the two energy landscapes (Fig. 4), is observed with deviation along the angle $\gamma$ (defined on $\hat{\tau}$-$\hat{n}$ plane with 0° being $\hat{\tau}$ and 90° being $\hat{n}$) being more energetically favored than that along $\theta$ (0° being $\hat{\chi}$ and 90° being $\hat{n}$). With the cases of colloidal rods along $\hat{n}, \hat{\chi}$, and $\hat{\tau}$ all giving distinct free energies, the biaxiality in the ensuing colloidal orientation probability distribution is explicit, which is attributed to the chirality of the host medium. The results are in agreement with the empirical evidence shown below.

## Experimental analysis of the colloidal orientations

For benchmarking purposes, we first analyze the orientational distribution of homeotropic disk dispersion in a chiral 5CB-based host, in which case the disk normals orient along $\hat{n}$ on average and the chirality-associated biaxiality is weaker than that of homeotropic rods orienting orthogonally to $\hat{n}$ as discussed below. Figure 5 demonstrates depth-resolved confocal photon-upconverting fluorescence micrographs of disk-shaped particles immersed in the chiral LC, where the frame $(\hat{n}, \hat{\chi}, \hat{\tau})$ is marked on each micrograph and the far-field structure is robustly controlled by surface boundary conditions on the confining substrates (Methods). The average normal direction of the colloidal disks in each confocal slice, which is parallel to $\hat{n}$, rotates along the sample depth, as shown with the edge-on perspective view (Fig. 5a). Subsequently, the twisted arrangement of the disk direction is analyzed and a twisting rate is found throughout the sample depth (Fig. 5b). The helical twist of molecular director $\hat{n}$ is identical to the one of colloidal orientations (Fig. 5b), and the period of the twisted arrangement of the colloids closely matches the molecular pitch $p$. Once the orientational distribution of the thin disks is projected onto the co-rotating cholesteric frame, we find Gaussian-like colloidal orientational distributions (Fig. 5c). With $\phi_c \approx 0.026\%$ direct interactions between colloidal disks are negligible[17] and each particle experiences an orientational potential imposed mainly by the surrounding molecular chiral LC. Accordingly, the 3D colloidal director distributions are weakly asymmetric due to the biaxiality imparted by the chiral molecular host, demonstrating different energy wells for the thin disk fluctuating in different angular directions. The peak widths (full width at half maximum, FWHM) of the colloidal orientation

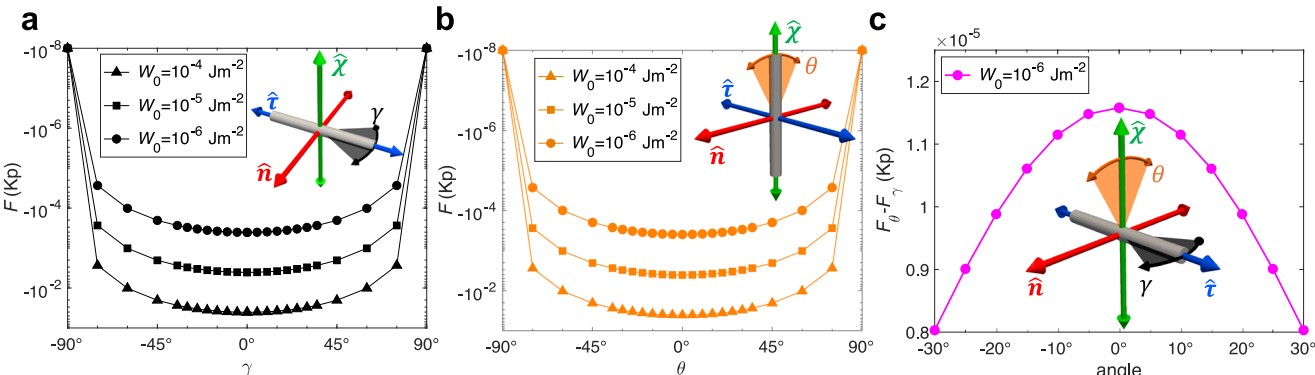

**Fig. 4 | Free energy landscape versus colloidal rod orientation. a** Computer-simulated free energy of chiral 5CB-based LC surrounding a homeotropic rod at different surface anchoring strengths $W_0$ and the azimuthal angle $\gamma$, defined as the angle on the $\hat{\tau}$-$\hat{n}$ plane in the inset. **b** Simulated free energy of rods with different values of polar angle $\theta$ on the $\hat{\chi}$-$\hat{n}$ plane. **c** The difference in energy profiles (**a**) and (**b**) of homeotropic rods rotated along $\hat{\tau}$ and $\hat{\chi}$ axis, as shown in the inset, simulated using anchoring strength $W_0 = 10^{-6}$ Jm⁻². The case of rods aligning along $\hat{n}$ (red axis) with the highest energy cost taken as an energy reference point for each value of $W_0$, while the free energy value is chosen to be $-10^{-8} Kp$ instead of 0 to avoid singularities when converting to a log-scale in (**a**) and (**b**). The axes in the insets define the frame colored as in Fig. 1. Cholesteric pitch $p = 30$ μm for all simulations and the average elasticity $K = 5.6$ pN for energy scaling.

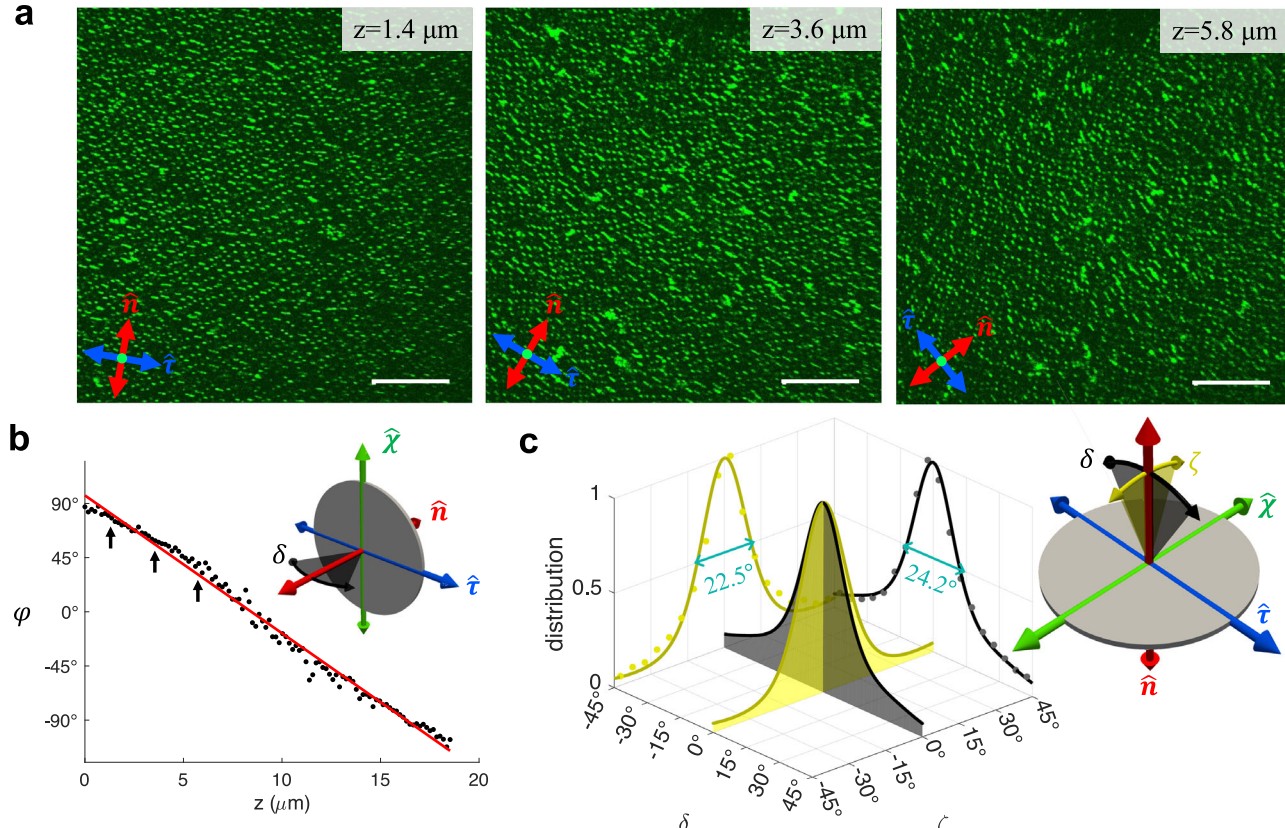

**Fig. 5 | Disk orientational biaxiality in a chiral hybrid LC. a** Photon-upconversion-based luminescence confocal images of homeotropic disks dispersed in chiral LC taken in planes perpendicular to $\hat{\chi}$ with the depths $z$ and directors ($\hat{\mathbf{n}}$ and $\hat{\tau}$) marked in each depth-resolved optical slice. **b** The azimuthal angle of the disk normal orientation $\varphi$ at different depth $z$ obtained for the same sample, with the slices corresponding to (**a**) indicated by arrows. Dots are the average angular values in each $z$ slice, and their linear fit is given by the red line, consistent with the LC pitch $p \approx 30\,\mu m$. The inset illustrates the observed orientational fluctuation of disks (black double arrow) in the frame ($\hat{\mathbf{n}}, \hat{\chi}, \hat{\tau}$). **c** Disk azimuthal orientational fluctuation $\delta = \varphi - qz$ and polar orientational fluctuation $\zeta = \pi/2 - \theta$ within the cholesteric frame (insets), with $\delta = 0$ or $\zeta = 0$ corresponding to the average disk normals' orientation along $\hat{\mathbf{n}}$ (red axis). Scale bars are 30 $\mu m$.

distributions slightly differ along two deviation angles (24.2° when changing $\delta$ and 22.5° for $\zeta$, see Fig. 5c), consistent with biaxial symmetry of an individual deformations-dressed disk with weak symmetry breaking for rotations around $\hat{\mathbf{n}}$, even though the particles themselves have uniaxial $D_\infty$ symmetry.

Contrasting the above weak-symmetry-breaking behavior, the same analysis reveals strong biaxial symmetry breaking for thin rods with perpendicular boundary conditions (Fig. 6). In agreement with numerical calculations (Fig. 4), the rods align on average along $\hat{\tau}$ axis in thermal equilibrium (Fig. 6a,b). After measuring the azimuthal angles $\varphi$ of the colloidal rod long axes in each confocal depth-resolved image slice and converting them to 3D distribution in the cholesteric frame, we clearly see a narrow distribution for the rod angle fluctuations (Fig. 6c). Furthermore, the orientational distributions of homeotropic rods are very different from those in non-chiral nematic hosts which feature a degeneracy of alignment along the $\hat{\chi}$ and $\hat{\tau}$-axes (Fig. 2c). By contrast, we find a strong energy well for rods deviating from $\hat{\tau}$ towards $\hat{\chi}$ (Fig. 6c). The hybrid LC system is thus strongly biaxial as the principal ordering directions for the molecular and colloidal particle axes are orthogonal. Besides, the distinct peak widths of the angular probability distributions demonstrate additional biaxial orientational order within the colloidal dispersion. We will show below that this biaxial symmetry breaking arises chiefly due to the energetic cost incurred by elastic distortions generated by such colloidal inclusions in the bulk of the chiral molecular host.

## Insights from analytical theory of LC-colloid interactions

We develop an analytical model to capture the main biaxiality-enabling energetic effects imparted onto the colloids by the chiral LC host, consistently with numerical simulations presented above. Based on our model describing colloidal surface anchoring energies as well as correction arising from elastic interactions (Methods), a simple Boltzmann distribution is proposed to predict the biaxial orientational probability of the colloidal rods and the phase diagram in Fig. 2 (see Supplementary).

The total external potential $F_{s,\text{tot}}$ acting on each colloid is expressed in terms of the Helmholtz free energy $F_s$ associated with the Rapini-Papoular (RP) surface anchoring forces for the undistorted host director plus a free energy contributions $\Delta F_{\text{dist}}$ originating from bulk elastic distortions (see Methods)

$$F_{s,\text{tot}} \sim F_s + \Delta F_{\text{dist}}. \tag{2}$$

From this we establish the orientational probability distribution $f(\hat{\mathbf{u}})$ through the Boltzmann distribution

$$f(\hat{\mathbf{u}}) \propto \exp(-\beta F_{s,\text{tot}}), \tag{3}$$

with $\beta^{-1} = k_B T$ the thermal energy in terms of Boltzmann's constant $k_B$ and temperature $T$. Since the distortion term cannot be resolved for any rod orientation but only for cases when the rod is aligned along either of the directions of ($\hat{\mathbf{n}}, \hat{\tau}, \hat{\chi}$) (see Method), we use the following

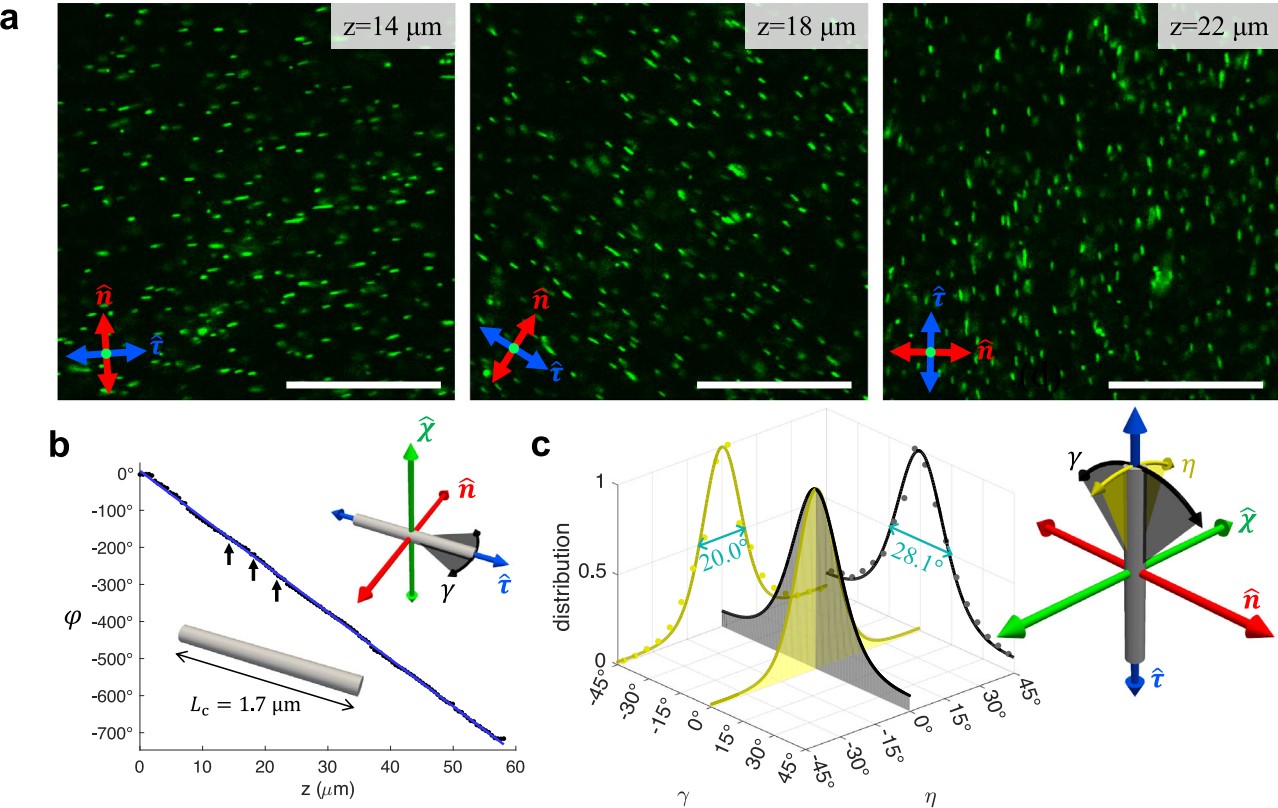

**Fig. 6 | Rod orientational biaxiality in the supercritical state. a** Depth slices of homeotropic rods dispersion in a chiral LC obtained using photon-upconversion-based luminescence confocal microscopy, in which the depth $z$ is measured along the $\hat{\chi}$ axis perpendicular to the micrographs. With a colloidal volume fraction $\phi_c \approx 0.001\%$ and host medium chirality $qL_c = 0.36$, this example corresponds to a supercritical biaxial colloidal fluid (Fig. 2). **b** The average orientation of the long axes of rod $\varphi$ in each depth $z$ slice (dots) and its linear fit (blue line), with the slices in (**a**) pointed out by arrows. LC pitch $p \approx 30\ \mu m$ and average rod length $L_c = 1.7\ \mu m$. **c** Orientational fluctuations of the rods measured in the cholesteric frame (inset), with the average direction being $\hat{\tau}$ (blue axis). All scale bars are 30 μm.

interpolation form

$$\Delta F_{dist}(\eta, \gamma) \sim \Delta F_{twist} \sin^2\eta + \Delta F_{tilt} \cos^2\eta \sin^2\gamma, \qquad (4)$$

in terms of the two angles $\eta = \theta - \frac{\pi}{2}$ and $\gamma = \delta - \frac{\pi}{2}$ represented in Fig. 6c and key elastic contributions; $\Delta F_{tilt} = F(\hat{\mathbf{u}} \parallel \hat{\mathbf{n}}) - F(\hat{\mathbf{u}} \parallel \hat{\tau})$ associated with tilting the rod away from the $\hat{\tau}$-axis towards the $\hat{\mathbf{n}}$-direction, discussed in Supplementary, and $\Delta F_{twist} = F(\hat{\mathbf{u}} \parallel \hat{\chi}) - F(\hat{\mathbf{u}} \parallel \hat{\tau})$ the energy cost associated with twisting the surface disclination wrapped along the body of the cylinder, detailed in Methods. From these analyses we established that $\Delta F_{twist}$ is a few tens of $k_BT$ (Fig. 7) whereas the elastic distortions due to tilting are much weaker ($\Delta F_{tilt} < k_BT$) and may, in fact, be neglected altogether for the weak anchoring regime considered in this study (Supplementary). The elastic energy is then minimal (zero) when the rods align along the $\hat{\tau}$ directions (with equilibrium angle $\theta^* = \pi/2$ and $\delta^* = \pi/2$) as observed in our experiments (Fig. 6). An overview of the orientational probability distributions associated with Eq. (2), based on the Boltzmann exponent Eq. (3), are depicted in Fig. 8 indicating that the rod preferentially aligns along the $\hat{\tau}$-axis with considerable orientational biaxiality developing around the main alignment direction.

Going back to the experimental case reported in Fig. 6 we may roughly estimate the energy contribution due to the twisted disclination from the width of the distributions depicted in panel **c**. For small angles $\eta$ the Boltzmann factor of Eq. (4) translates into a simple Gaussian distribution

$$f(\eta) \propto \exp(-\Delta F_{twist}\eta^2), \qquad (5)$$

and we identify a standard Gaussian FWHM $= 1.67/\sqrt{\Delta F_{twist}}$. This subsequently gives $\Delta F_{twist} \approx 22.1 k_BT$ for homeotropic rods suggesting that the thermal motion of the rods is assuredly insufficient to overcome the energy difference between the $\hat{\mathbf{u}} \parallel \hat{\tau}$ and $\hat{\mathbf{u}} \parallel \hat{\chi}$ alignment directions.

## Colloidal orientational order parameters

To quantify the biaxial orientational symmetry-breaking of the colloids observed in our experiment, we measured the order parameters. Following standard definitions, we express the orientational order of the colloids through a local mean-field tensorial order parameter[17,38] spanned by the orthogonal cholesteric frame depicted in Fig. 1

$$\mathbf{Q}^{(c)} = S_c \left(\frac{3}{2}\hat{\mathbf{n}} \otimes \hat{\mathbf{n}} - \frac{\mathbf{I}}{2}\right) + \frac{\Delta_c}{2}\left(\hat{\tau} \otimes \hat{\tau} - \hat{\chi} \otimes \hat{\chi}\right). \qquad (6)$$

The colloidal $S_c$ scalar order parameter can be extracted from the orientational distribution $f$ via

$$S_c = \langle \mathcal{P}_2(\hat{\mathbf{u}} \cdot \hat{\mathbf{n}})\rangle_f, \qquad (7)$$

with $\hat{\mathbf{u}}$ denoting the $C_\infty$ symmetry axis of cylindrical particles and $\langle \ldots \rangle_f$ an ensemble average. Similarly, we extract the $\Delta_c$ order parameter that quantifies the relative orientational order with respect to the principal directions orthogonal to $\hat{\mathbf{n}}$

$$\Delta_c = \frac{1}{2}\langle(\hat{\mathbf{u}} \cdot \hat{\tau})^2 - (\hat{\mathbf{u}} \cdot \hat{\chi})^2\rangle_f. \qquad (8)$$

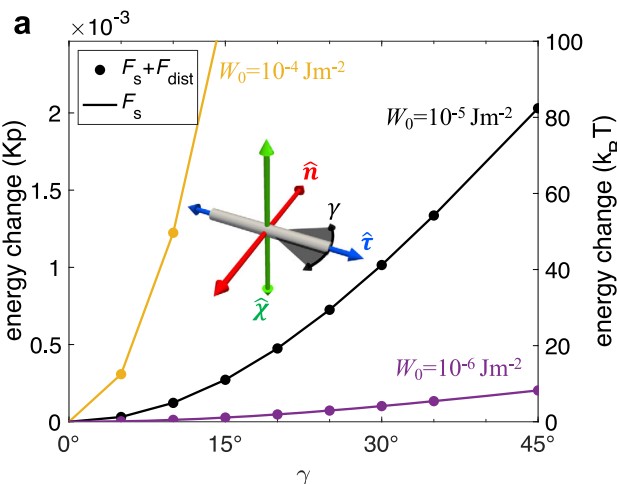
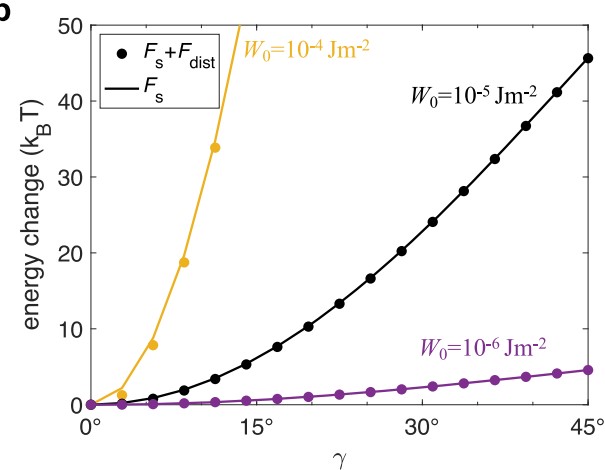

**Fig. 7 | Energy landscape for colloidal rod reorientation. a** Numerical LC energies for the dispersion of homeotropic rods at different angles $\gamma$ defined in the inset. The lines illustrate pure surface energy contribution and the dots include elastic distortion energies. **b** The corresponding theoretical values with elastic distortion (Eq. (4)) or without (Eq. (23)). Surface anchoring strengths $W_0$ are marked for each data set. Molecular cholesteric pitch $p = 30\,\mu m$, average elastic constant $K = 5.6\,pN$, rod length $L_c = 1.7\,\mu m$ and diameter $D_c = 28\,nm$ for all simulations and calculations. The energy zero points are chosen at $\gamma = 0$ for clarity.

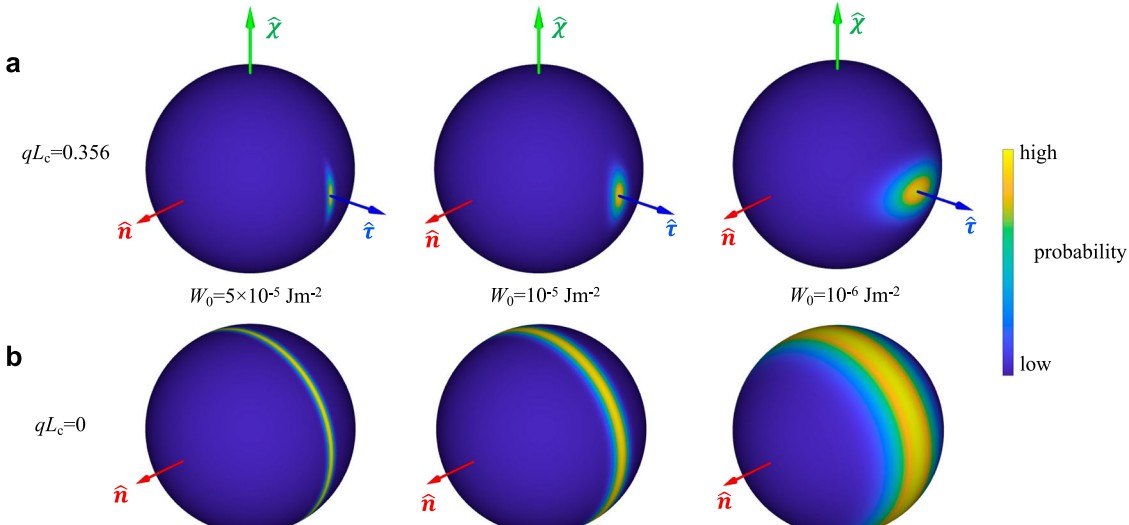

**Fig. 8 | Rod orientational probability in hybrid LCs.** Unit-sphere projections of the predicted orientational distribution of thin rods dispersed in (**a**, top row) chiral or (**b**, bottom row) non-chiral LCs. Empirical values of cholesteric pitch $p = 30\,\mu m$ and $\Delta F_{twist} = 22.1 k_B T$ are used for the chiral cases in (**a**) with $L_c = 1.7\,\mu m$ and $D_c = 28\,nm$ used for all calculations. The surface anchoring coefficients are marked for each column.

Results are gathered in Table 1. $S_c$ represents the strength of orientational confinement (Eq. (7)) which depends principally on the surface-anchoring properties of the colloid surface and the shape and dimensions of the colloids. The very strong biaxial symmetry breaking observed for homeotropic rods is clearly reflected by the values of $\Delta_c$. Since the order parameters are computed within the cholesteric reference frame, the negative values of $S_c$ reflect the case in which the average colloidal director lies orthogonal to $\hat{n}$.

Here the biaxiality of our hybrid molecular-colloidal system $\Delta_c$ is measured relative to the director of the host medium $\hat{n}$. On the other hand, one could also focus on the biaxiality of the colloidal subsystem alone and quantify the biaxiality in the colloidal orientational distributions. For homeotropic rods, the orientational biaxiality can be quantified as $\frac{1}{2}\langle(\hat{u}\cdot\hat{n})^2 - (\hat{u}\cdot\hat{\chi})^2\rangle_f$ in contrast to Eq. (8), and the value is found to be 0.12 in our experiment. Figure 6c illustrates such colloidal distribution biaxiality of homeotropic rods by

visualizing the fluctuation of rod axis around $\hat{\tau}$ axis which is related to the difference in peak widths on the two orthogonal planes containing $\hat{\tau}$. Furthermore, Fig. 8 shows that colloidal orientational distribution biaxiality can be controlled by tuning the interplay of surface anchoring and elastic energies associated with incorporating the particle into the LC.

## Colloidal shape matters

While it has been known that nonchiral nematic LCs can impart orientational order on anisotropic particles dispersed within them, the symmetry of the ensuing LC colloidal system always remained the same as that of the uniaxial host medium, even if the particles had lower symmetry[10,39,40], unless the concentration of colloidal conclusions approached the values at which direct inter-particle interactions could lead to the emergence of lower-symmetry LC-colloidal composites like biaxial orthorhombic nematic fluids[17,18]. An unexpected

**Table 1 | Uniaxial ($S_c$) and biaxial ($\Delta_c$) colloidal order parameters measured in the cholesteric frame for experiments shown in Fig. 5 and Fig. 6**

| Sample | $S_c$ | $\Delta_c$ |
|---|---|---|
| Homeotropic disks | 0.66 | 0.067 |
| Homeotropic rods | -0.28 | 0.79 |

finding presented above is that the chiral nematic LC host medium imparts biaxial orientational order even on high-symmetry uniaxial particles like rods and disks, which we have demonstrated is based on the minimization of the overall energetic costs associated with dispersing colloidal inclusions in the LC host medium with well-defined alignment of the far-field helical axis and helicoidal director configurations. As the particle shape and boundary conditions play important roles in defining interactions between the LC host medium and colloidal inclusions, we expect even richer phenomena and new soft condensed matter phases to emerge for other colloidal shapes of low symmetry, like, for example, bent-rod particles[41].

Details of the geometric shape play important roles even in defining behavior of uniaxial-symmetry particles with the same surface boundary conditions. Comparing colloidal disks and rods immersed in a chiral LC, we clearly observe that the biaxial order developed at the level of the colloids is most pronounced for rods, whose energy-favored orientation is along $\hat{\boldsymbol{\tau}}$ and perpendicular to $\hat{\mathbf{n}}$ (Fig. 3). This symmetry breaking at the single particle level, where the symmetry axis of the colloidal rod is orthogonal to $\hat{\mathbf{n}}$, is more striking than for colloidal disks aligned with their symmetry axis along $\hat{\mathbf{n}}$, consistent with our analytical theory and numerical modeling (Fig. 8). However, even for disks the enhanced, emergent biaxiality goes far beyond the previous theoretical predictions[19] for a single-component molecular LC. To illustrate this point for the weak molecular chirality regime we characterize the leading order contribution of chirality to colloidal biaxiality by expanding the biaxial order parameter up to the quadratic order in $q = 2\pi/p$

$$\Delta_c = \Delta_0 (qD_c)^2 + \mathcal{O}[(qD_c)^4] \qquad (9)$$

while following the analytical energetic analysis above, where $\Delta_0$ is of $\mathcal{O}(1)$. Even for homeotropic colloidal disks the measured $\Delta_c$ (Table 1) and its predicted values of the order of 0.1 are many orders of magnitude larger than what one would expect ($10^{-7}$) for a purely molecular chiral nematic of the same pitch, as estimated based on the $(qL_m)^2$ scaling[19]. Colloidal homeotropic rods exhibit even much stronger biaxial order and unexpected emergent behavior far exceeding the chirality-induced biaxial symmetry breaking of purely molecular cholesterics (Fig. 2f), as indicated by the values in Table 1 and displayed in Fig. 2. Even if biaxiality at the colloidal particle subsystem level could be neglected (say when the surface anchoring and elastic energy interplay illustrated in Fig. 8a would make the distribution of rods roughly uniaxial relative to the average direction of ordering along $\hat{\boldsymbol{\tau}}$), the overall hybrid system would still be biaxial simply because the molecular and colloidal rods order mutually orthogonal to each other (and one cannot approximate the symmetry of two mutually orthogonal cylinders with that of just one). Furthermore, in view of the role played by elastic distortions around the colloidal rods, tuning the elastic properties of the background molecular LC provides a means to control the emergent biaxiality of our hybrid LC, in addition to the dimensions and surface chemistry of the colloids and the degree of molecular chirality. Having such ability to control and design biaxiality of orientational ordering is not possible for purely molecular, colloidal or micellar systems where biaxiality has been studied so far[14–18].

While in this study we focus on just several out of many possible examples of emergent long-range biaxial nematic order in hybrid molecular-colloidal systems, many additional scenarios are discussed elsewhere in our future publication, revealing how this emergent behavior can be controlled by changing the boundary conditions on the colloidal particles, their geometric shapes, density regimes, and so on.

## Discussion

We have demonstrated that immersing uniaxial, non-chiral colloidal rods and disks into a low-molecular-weight cholesteric LC host leads to emergent biaxial order that we identify at colloidal, molecular and composite system levels by combining experiment with numerical simulation and analytical theory (Fig. 9). Unlike the previously studied case of hybrid molecular-colloidal biaxial phases, we observe multi-level biaxial symmetry-breaking at ultralow colloidal content where colloid-colloid interactions are negligible. We uncover a highly unconventional scenario in which the hybrid molecular-colloidal nature of the LC mixture along with the chirality of the molecular host generates strongly biaxial orientation order resulting in long-range ordered biaxial colloidal supercritical fluid states. Unlike in non-chiral hybrid molecular-colloidal LCs, where biaxial order emerges only at modest to high volume fractions of the anisotropic colloidal particles, above a uniaxial-biaxial transition critical concentration[17,18], the orientational probability distribution of colloidal inclusions immersed in chiral nematic hosts are unavoidably biaxial even at vanishingly low particle volume fractions.

A particularly striking manifestation of biaxial symmetry-breaking is encountered for cholesterics doped with colloidal rods. Driven by a combination of surface anchoring forces and an energy penalty incurred by twisting a weakly developed surface disclination along the rod main axis, the rods have a strong tendency to align perpendicular to both the helical axis and the local cholesteric director, thus imparting a two-fold $D_{2h}$ orientational symmetry onto the hybrid system at each point along the cholesteric helix. By means of numerical minimization of the free energy, we revealed that the emergent biaxiality in our systems manifests already at ultralow colloidal concentrations, and we find consistent agreement between our modeling predictions and the experimental observations.

Our results pave the way towards controlled biaxial order in soft matter. By harnessing the interplay of chiral and biaxial symmetries, future research efforts could be directed along the following emergent avenues. At larger colloidal concentrations a richer phenomenology could be expected and explored due to the more prominent roles expected to be played by steric, electrostatic or defect-mediated colloid-colloid interactions further enriching the surface anchoring and elastic forces discussed here. Besides the emergent symmetry breaking, one could, in principle, also apply electric or magnetic fields to reconfigure either molecular or colloidal sub-systems, or both, to achieve even lower externally induced symmetries of LCs, for instance, corresponding to triclinic or monoclinic point groups. Finally, by realizing topological solitons in the molecular-colloidal hybrid system with nontrivial chirality and biaxiality, one could probe the stability of topological structures for various low-symmetry order parameter spaces. While ferromagnetic colloidal particle dispersions have already provided insight into the possibility of formation of solitons in polar chiral liquid crystals[42], this study could be extended to symmetries differing from nonpolar and polar uniaxial LCs, for example, by exploring multi-dimensional solitonic structures corresponding to the $SO(3)/D_2$ order parameter space.

## Methods

We study a soft matter system that has a helicoidal configuration with a cholesteric frame ($\hat{\mathbf{n}}, \hat{\boldsymbol{\chi}}, \hat{\boldsymbol{\tau}}$) composed of three mutually perpendicular directors/axes and a helical pitch $p$ (Fig. 1), which is hardly perturbed by the introduction of thin colloidal disks or rods with weak surface anchoring boundary conditions. The colloidal particles we used are all

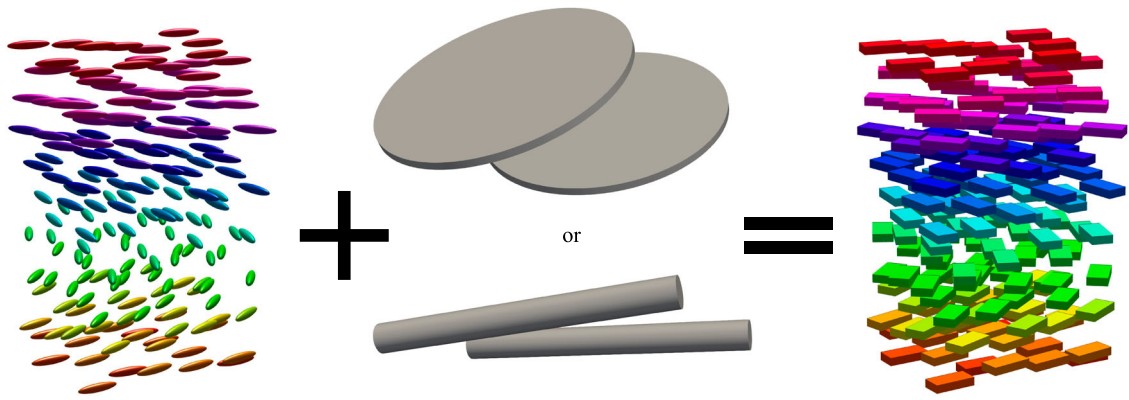

**Fig. 9 | Local orthorhombic biaxial symmetry breaking in a chiral hybrid LC.** Our molecular-colloidal hybrid system with emergent biaxial symmetry consists of purely uniaxial building blocks. The chirality effects at different scales yield an effective behavior of a biaxial chiral molecular-colloidal LC.

uniaxial of shape with a large aspect ratio. Their orientations relative to the $(\hat{\mathbf{n}}, \hat{\chi}, \hat{\tau})$ frame depend on their geometry and surface boundary conditions that are controlled via chemical treatment, as described below. We will refer to these colloidal objects as homeotropic rods or disks because the surface-energy-minimizing boundary conditions of the particles promote the orientation of the LC molecules to be locally perpendicular to the colloid surface.

### Synthesis of colloidal disks and rods
Disk or rod-shaped $\beta - NaYF_4$:Yb/Er particles are synthesized following the hydrothermal synthesis methods described in detail elsewhere[17,18,43–45]. Precursors and solvents used for the synthesis of colloidal particles are of analytical grade and used without additional purifications, and they are bought from Sigma Aldrich if not specified otherwise. To synthesize nanodisks, 0.7 g of sodium hydroxide (purchased from Alfa Aesar) is dissolved in 10 ml of deionized water and then added with 5 ml of oxalic acid solution (2g, 19.2 mmol) at room temperature to obtain a transparent solution. Under vigorous stirring, we then add 5 ml of sodium fluoride solution (202 mg, 4.8 mmol) to the mixture. After 15 min of stirring, 1.1 ml of $Y(NO_3)_3$ (0.88 mmol), 0.35 ml of $Yb(NO_3)_3$ and 0.05 ml of $Er(NO_3)_3$ are added into the mixture while the stirring continues for another 20 min at room temperature. Subsequently, the solution is transferred to a 40-ml Teflon chamber (Col-Int. Tech.) and heated to and kept at 200 °C for 12 h. The mixture is then cooled down naturally to room temperature, and the particles precipitated at the bottom are collected by centrifugation, rinsed with deionized water multiple times, and finally dispersed in 10 ml of deionized water. Colloidal rods are prepared using a similar protocol: 1.2 g of NaOH is dissolved in 5 ml of deionized water and mixed with 7 ml of ethanol and 20 ml of oleic acid under stirring, followed by adding 8ml of NaF (1 M), 950 μl of $Y(NO_3)_3$ (0.5 M), 225 μl of $Yb(NO_3)_3$ (0.2 M), and 50 μl of $Er(NO_3)_3$ (0.2 M) and stirring for 20 min. The obtained white viscous mixture is transferred into a 50 ml Teflon chamber, kept at 190 °C for 24 h, and then cooled down to room temperature. The particles deposited at the bottom of the Teflon chamber are collected and washed with ethanol and deionized water multiple times and finally dispersed in cyclohexane.

### Surface functionalization of the colloidal particles
Homeotropic (perpendicular) surface anchoring boundary conditions for the director of 5CB (pentylcyanobiphenyl or 4-cyano-4'-pentylbiphenyl) molecules on the $\beta$-NaYF$_4$:Yb/Er disk surfaces is controlled through surface-functionalization with a thin layer of silica and polyethylene glycol. First, 5 ml of hydrogen peroxide ($H_2O_2$) is added to 1 ml of colloidal disk dispersion in deionized water. Then, under vigorous mechanical agitation, 100 μl of nitric acid is added drop by drop into the solution. After 12 h of agitation, disks are separated from the

liquid by centrifugation and transferred into 1 ml of ethanol. The colloidal dispersion is then mixed with 75 mg of polyvinyl pyrrolidone (molecular weight 40,000) in 4 ml of ethanol and kept under continuous mechanical agitation for another 24 h. The particles are collected and redispersed in 5 ml of ethanol, before the addition of 200 μl of ammonia solution and 6 μl of tetraethyl orthosilicate under mechanical agitation that lasts 12 h. Disks are collected, washed with ethanol and deionized water, and redispersed in 4 ml of ethanol. The pH value of the mixture is adjusted to 12 by adding ammonia solution (28% in water). Then, under mechanical agitation at 35 °C, we add 35 mg of silane-terminated polyethylene glycol (molecular weight 5,000, dissolved in 1 ml of ethanol at 50 °C) to the solution. After another 12 h of agitation, the surface-functionalized disks are again collected, washed with ethanol and water, and dispersed in 1 ml of ethanol.

As for the hydrothermal-synthesized rods, the surface chemical treatment not only provides the desired anchoring preference but also controls the cylinder aspect ratio. For this, 4 ml of the nanorod dispersion is added with 200 μl of HCl in 2 ml of water and kept stirred overnight. The nanorods are then transferred from organic to aqueous phases. The nanorods are collected by centrifugation, washed with deionized water and ethanol three times, dispersed in deionized water, and then finally re-dispersed in ethanol. The process of etching with acid and redispersion is repeated two more times, with HCl treatments of 12 h and 3 h, respectively. The aspect ratio of the nanorods is increased during acid treatment to an average value of $L_c/D_c \approx 60$.

### Particle dispersion in molecular host and sample preparation
A small amount of left-handed chiral dopant cholesterol pelargonate is added into molecular 5CB (Frinton Labs and Chengzhi Yonghua Display Materials Co. Ltd). To obtain the equilibrium pitch $p$ of discrete values 10, 15, 20, 30 and 60 μm of the molecular chiral LC mixtures, the weight fraction of the used chiral additive is roughly estimated by $c_d = \frac{1}{6.25p}$. The actual pitch is later revealed using optical microscopy by observing the periodicity of defect lines in Gradjean-Cano wedge cells[46]. The surface-functionalized particles are then dispersed into such molecular chiral LC. In a typical experiment, 20 μl of colloidal dispersion in ethanol is mixed with 20 μl of the molecular LC. The mixture is then heated to 75 °C and kept for 2 h to completely evaporate the organic solvent. A well-dispersed colloidal-molecular hybrid LC is usually obtained after quenching back to room temperature under mechanical agitation[33,47,48]. Additional centrifugation can be carried out to remove the particle aggregation formed during the isotropic to chiral nematic phase transition of the molecular LC. Hybrid LCs containing the colloidal dispersion are infiltrated into glass cells with gap thickness typically chosen to be between $p/2$ and $10p$, which is experimentally set using Mylar films or silica spheres. Cell

substrates are coated with 1wt.% aqueous polyvinyl alcohol and rubbed unidirectionally so that parallel boundary conditions for 5CB molecules are formed at confining glass surfaces, and that the helical axis $\hat{\chi}$ is perpendicular to the glass substrates.

## Microscopy and characterization of colloidal orientations

We use different optical microscopy methods to visualize the colloidal orientations inside the hybrid LC, among which are three-photon excitation fluorescence polarizing microscopy (3PEFPM), photon-upconverting confocal microscopy and polarizing optical microscopy[17,18]. Using 3PEFPM, optical imaging of director structures of the molecular host medium is performed using a multimodal 3-dimensional (3D) nonlinear imaging system built around a confocal system FV300 (Olympus) and an inverted microscope (Olympus IX-81)[47,49]. The 3D imaging of the $\beta$-NaYF$_4$:Yb/Er particles designed to exhibit upconversion luminescence is performed with the same setup when the colloidal dispersions are excited with a laser light at 980 nm; this photon-upconversion-based imaging of colloidal particles minimizes the background signal from the molecular LC, making such a technique ideal for our study. A 100 × objective (Olympus UPlanFL, numerical aperture 1.4) and a 980-nm pulsed output from a Ti:Sapphire oscillator (80 MHz, Coherent, Chameleon ultra) are utilized, along with a set of Galvano mirrors on the optical path to achieve sufficient positional accuracy while scanning the sample horizontally. In addition, the vertical re-positioning is achieved by a stepper motor on which the objective could be adjusted to focus at the desired sample depth, enabling 3D scanning with high accuracy. Luminescence signals are epi-collected using the same objective before being sent through a pinhole and detected by a photomultiplier tube. The data obtained from several scanning planes are combined into a 3D tiff image to be analyzed at a later time.

The colloidal orientations, representing the normal direction of the disk or the long axis of the rod, are analyzed based on its projections on multiple two-dimensional (2D) optical slices of a 3D sample using ImageJ software (freeware from the National Institute of Health,[50]). The ensuing data are transferred to Matlab software for visualization as well as for further analysis. The contrast and brightness of the images are carefully adjusted to avoid the interference of colloids out of focus. From the 3D stacks of images, the $\hat{n}$-$\hat{\tau}$ slice plane perpendicular to the helical axis $\hat{\chi}$ gives the azimuthal orientational distribution ($\varphi$), whereas the vertical slice plane parallel to $\hat{\chi}$ reveals the polar distribution ($\theta$) of colloidal orientations. We follow the methodology described in detail in refs. 16–18 and perform a complete analysis of angles describing the colloidal orientations relative to the helical far-field background by imaging the entire sample volume. After the analysis of particles by ImageJ, average azimuthal colloidal orientations are calculated for the data obtained in each $\hat{n} - \hat{\tau}$ slice plane and plotted against the sample depth ($z$) position of the cross-sectional plane, revealing the helical twist of the colloidal axes. The corresponding helical pitch $p$ of each 3D volume is subsequently calculated from the slope of the linear dependence of the azimuthal angle on the vertical position ($d\varphi/dz = q = 360°/p$) and is in agreement with the initially designed value mentioned above, confirming the undisturbed molecular helical pitch at relatively low colloidal concentrations. Finally, the colloidal orientation distribution is visualized in the cholesteric frame as follows: the azimuthal angle in the molecular coordinate is calculated by subtracting the molecular twist from the measured colloidal orientations, $\delta = \varphi - qz$ representing the fluctuation of colloidal orientation around that of a perfect helix. The nonorientable property of the colloidal axis $\hat{u} = -\hat{u}$ enabled us to express the fluctuation angles, $\delta = \delta + \pi$, for example, within a [-90°,90°] range. Histograms of the angular probability distribution with 5° bin width are calculated for each fluctuation angle, and numerical fitting based on the theoretical model is performed to each distribution accordingly and the peak width is quantified at half peak height (full width at half

maximum, FWHM). The fit function is justified from the analytical prediction for the angular dependence of the surface anchoring energy (Eq. (20) and Eq. (23) below) and the elastic energy Eq. (4) discussed in the sections below. The visualization of angle distributions is cropped to a smaller angle range after calculation performed in the full [-90°,90°] range. The histogram data sets are subsequently utilized in the computation of the colloidal orientation order parameters, as summarized in the Results.

## Computer simulation of the molecular host surrounding the colloids

Computer simulations are carried out to study the interplay between molecular LC order near the colloidal surface and the colloidal orientation. For each numerical computation with a single particle surrounded by the molecular LC host, we calculate the distortion and realignment of the host medium induced by the surface anchoring effect at the particle surface. Therefore, the dimensions of colloidal particles, along with their anchoring types, are represented as boundary conditions inside the numerical volume for the host medium and are kept constant for each simulation[17,51,52]. Specifically, cylindrical rods with width $D_c = 28$ nm and length $L_c = 1.7$ μm are adopted with homeotropic boundaries (molecular director $\hat{n}$ perpendicular to surface), if not specified otherwise.

We then minimize the mean-field Landau-de Gennes free energy for the molecular LC host, including bulk and surface energy terms, during which the host medium finds the lowest-energy configuration to accommodate the introduction of particles. The bulk energy density consists of a thermotropic bulk free energy density describing the isotropic-nematic transition of LCs complemented with elastic contributions associated with LC director distortions occurring in the nematic bulk[5,17,51–53]:

$$f_{\text{bulk}}^{\text{LC}} = \frac{A}{2}\mathbf{Q}_{ij}^{(m)}\mathbf{Q}_{ji}^{(m)} + \frac{B}{3}\mathbf{Q}_{ij}^{(m)}\mathbf{Q}_{jk}^{(m)}\mathbf{Q}_{ki}^{(m)} + \frac{C}{4}\left(\mathbf{Q}_{ij}^{(m)}\mathbf{Q}_{ji}^{(m)}\right)^2$$
$$+ \frac{L_1}{2}\left(\frac{\partial \mathbf{Q}_{ij}^{(m)}}{\partial x_k}\right)^2 + \frac{L_2}{2}\frac{\partial \mathbf{Q}_{ij}^{(m)}}{\partial x_j}\frac{\partial \mathbf{Q}_{ik}^{(m)}}{\partial x_k}$$
$$+ \frac{L_3}{2}\frac{\partial \mathbf{Q}_{ij}^{(m)}}{\partial x_k}\frac{\partial \mathbf{Q}_{ik}^{(m)}}{\partial x_j} + \frac{L_4}{2}\epsilon_{ijk}\mathbf{Q}_{il}^{(m)}\frac{\partial \mathbf{Q}_{kl}^{(m)}}{\partial x_j}$$
$$+ \frac{L_6}{2}\mathbf{Q}_{ij}^{(m)}\frac{\partial \mathbf{Q}_{kl}^{(m)}}{\partial x_i}\frac{\partial \mathbf{Q}_{kl}^{(m)}}{\partial x_j},$$

$$(10)$$

with the 3-by-3 matrix $\mathbf{Q}^{(m)}$ being the molecular tensorial order parameter describing the local average molecular ordering, $x_i$ ($i = 1$–3) being cartesian coordinates, and $\epsilon$ the 3D Levi-Civita tensor. Summation over all indices is implied. Among the bulk energy terms, $A$, $B$, and $C$ are thermotropic constants and $L_i$ ($i = 1$–4, 6) are the elastic constants related to the Frank-Oseen elasticities via

$$L_1 = \frac{2}{27\left(S_{\text{eq}}^{(m)}\right)^2}(K_{33} - K_{11} + 3K_{22})$$
$$L_2 = \frac{4}{9\left(S_{\text{eq}}^{(m)}\right)^2}(K_{11} - K_{24})$$
$$L_3 = \frac{4}{9\left(S_{\text{eq}}^{(m)}\right)^2}(K_{24} - K_{22})$$
$$L_4 = \frac{8}{9\left(S_{\text{eq}}^{(m)}\right)^2}K_{22}\frac{2\pi}{p}$$
$$L_6 = \frac{4}{27\left(S_{\text{eq}}^{(m)}\right)^3}(K_{33} - K_{11}),$$

$$(11)$$

with $K_{11}$, $K_{22}$, $K_{33}$ and $K_{24}$ respectively denoting the splay, twist, bend and saddle-splay elastic moduli, and $S_{eq}^{(m)}$ being the equilibrium uniaxial scalar order parameter. On the other hand, the contribution due to the boundary condition of the molecular LC at the colloidal surfaces reads

$$f_{surf}^{LC} = W_0 \left( P_{ik} \tilde{Q}_{kl} P_{lj} - \frac{3}{2} S_{eq}^{(m)} \cos^2 \theta_e P_{ij} \right)^2, \quad (12)$$

with $W_0$ the surface anchoring strength, $\mathbf{P} = \hat{\mathbf{v}} \otimes \hat{\mathbf{v}}$ the surface projection tensor, $\hat{\mathbf{v}}$ the surface normal director, and $\tilde{\mathbf{Q}} = \mathbf{Q}^{(m)} + \frac{1}{2} S_{eq}^{(m)} \mathbf{I}$. The equilibrium angle $\theta_e = 0$ corresponds to vertical or homeotropic anchoring at the boundary[54].

The total free energy of molecular LC is numerically minimized based on the forward Euler method integrating

$$\frac{d\mathbf{Q}^{(m)}}{dt} = -\frac{dF_{total}^{LC}}{d\mathbf{Q}^{(m)}}, \quad (13)$$

with $t$ being the scaled energy-relaxation time of the LC. Adaptive Runge-Kutta method (ARK23) and FIRE, Fast Inertial Relaxation Engine, are adopted to increase numerical efficiency and stability[55,56]. In each computation iteration, the total free energy is given by the integration of a bulk energy density $f_{bulk}^{LC}$ over LC volume and a surface one $f_{surf}^{LC}$ over colloid-molecule interfaces, with the colloidal volume excluded in the integral of free energy densities.

The simulations are carried out in a Cartesian colloidal frame using equidistant grid sets. The steady-state and termination of simulation are determined by the change in total free energy in each numerical iteration, which is usually monotonic decreasing. The director $\hat{\mathbf{n}}$ realignment is identified by comparing final energy-minimizing molecular director structures to the initial ones with uniform helices, and total free energies are compared for various chosen colloidal alignment angles. The following parameters are used for all computer simulations: $A = -1.72 \times 10^5 \mathrm{Jm^{-3}}$, $B = -2.12 \times 10^6$ $\mathrm{Jm^{-3}}$, $C = 1.73 \times 10^6$ $\mathrm{Jm^{-3}}$, $K_{11} = 6$ pN, $K_{22} = 3$ pN, $K_{33} = 10$ pN, $K_{24} = 3$ pN and $S_{eq}^{(m)} = 0.533$[17,51]. A home-built Matlab code can be found in ref. 17, and more details regarding Landau-de Gennes simulation of molecular LCs can be found, for example, in refs. 51,56.

**Analytical theory**

We start with a simple model for pure surface anchoring in the absence of weak elastic distortions. While such a simplified description suffices to explain our experimental observations for disks, it does not account for the preferred alignment direction of colloidal rods immersed in a chiral molecular LC. We then proceed with quantifying director distortions around the rods and the impact of chiral twist on the elastic energy of the LC host incurred by a single thin colloidal rod.

**Surface anchoring free energy of a thin cylindrical disk immersed in a cholesteric host.** We consider a chiral LC with a director field $\hat{\mathbf{n}}_h(z)$ twisted along the $\hat{\boldsymbol{\chi}}$-axis of a Cartesian laboratory frame that we denote by the normalized unit vectors $(\hat{\mathbf{x}}, \hat{\mathbf{y}}, \hat{\mathbf{z}})$ where $\hat{\mathbf{z}}$ coincides with the helical axis $\hat{\boldsymbol{\chi}}$ in Fig. 1. The helical director field of a cholesteric, denoted by subscript $h$, may be parameterized as follows

$$\hat{\mathbf{n}}_h(z) = \hat{\mathbf{x}} \cos qz + \hat{\mathbf{y}} \sin qz, \quad (14)$$

in terms of the cholesteric pitch $p = 2\pi/q$ and handedness $q < 0$ that we assume left-handed in agreement with experimental reality without loss of generality. Next, we immerse an infinitely thin cylindrical disk with aspect ratio $D_c/L_c \to \infty$ into a cholesteric host. The presence of the colloid will generate elastic distortions of the uniform director field $\hat{\mathbf{n}}_h(\mathbf{r})$ due to the specific anchoring of the molecules at the colloidal surface, quantified by the surface

anchoring strength $W_0 > 0$ (units energy per surface area). The extent of the elastic distortions around the colloid surface depends on the surface extrapolation length $\ell_s = K/W_0$ where $K$ denotes the average elastic constant of the thermotropic liquid crystal[57]. In the first part of our analysis, we focus on the regime of infinitely large surface extrapolation length ($\ell_s \to \infty$), in which case the elastic distortions around the immersed colloid are absent. For finite $\ell_s$, such as in the experiments, elastic distortions are weak but non-negligible. While we shall ignore the impact of these weak director distortions for colloidal disks, we will quantify them in detail for the case of colloidal rods as they turn out to be of crucial importance in explaining the experimental observations in Fig. 6.

If we assume the molecular director field $\hat{\mathbf{n}}$ to be given by Eq. (14), we can obtain the surface anchoring free energy from the Rapini-Papoular (RP) model by integrating over the colloid surface denoted by $\mathcal{S}$[58,59]

$$F_s = -\frac{1}{2} W_0 \oint d\mathcal{S} (\hat{\mathbf{n}}_h \cdot \hat{\mathbf{v}}(\mathcal{S}))^2, \quad (15)$$

where $\hat{\mathbf{v}}$ is a unit vector specifying the preferred direction of LC alignment at the colloid surface. In case of disks with uniform homeotropic anchoring across the surface gives $\hat{\mathbf{v}}(\mathcal{S})$ is equal to the disk normal vector $\hat{\mathbf{u}}$. Ignoring rim effects which are deemed irrelevant for the thin disks considered in experiment $L_c \ll D_c$ and defining the orthonormal vectors $\hat{\mathbf{e}}_{1,2}$ perpendicular to $\hat{\mathbf{u}}$ we parameterize the surface of the circular face of the disk as follows

$$\mathbf{r}_{\mathcal{S}} = \mathbf{r}_0 + t \left[ \hat{\mathbf{e}}_1 \sin \xi + \hat{\mathbf{e}}_2 \cos \xi \right], \quad (16)$$

with vector $\mathbf{r}_0$ locating the center of the circular face while $0 < t < D_c/2$ denoting the radial distance away from the center and $0 < \xi < 2\pi$ the azimuthal direction on the disk surface. The main symmetry axis of the colloidal disk is parameterized in the lab frame as $\hat{\mathbf{u}} = u_x \hat{\mathbf{x}} + u_y \hat{\mathbf{y}} + u_z \hat{\mathbf{z}}$ with $(u_x, u_y, u_z) = (\sin \theta \sin \varphi, \sin \theta \cos \varphi, \cos \theta)$ in terms of a polar $\theta$ and azimuthal angle $\varphi$ with respect to the helical axis $\hat{\mathbf{z}} = \hat{\boldsymbol{\chi}}$. Based on this two auxiliary unit vectors maye be defined from $\hat{\mathbf{e}}_1 = (\hat{\mathbf{x}} u_y - \hat{\mathbf{y}} u_x)/|\sin \theta|$ and $\hat{\mathbf{e}}_2 = \hat{\mathbf{u}} \times \hat{\mathbf{e}}_1$ so that the $(\hat{\mathbf{u}}, \hat{\mathbf{e}}_1, \hat{\mathbf{e}}_2)$ constitutes an orthonormal particle-based Cartesian frame. Switching to the new coordinate frame we write $\oint d\mathcal{S} = \int_0^{2\pi} d\xi \int_0^{D_c/2} dtt = \pi D_c^2/4$ and the surface anchoring energy per disk face can be re-expressed as

$$F_s = -\frac{1}{2} W_0 \int_0^{2\pi} d\xi \int_0^{\frac{D_c}{2}} dtt [\hat{\mathbf{n}}_h (\Delta \mathbf{r}_{\mathcal{S}} \cdot \hat{\boldsymbol{\chi}}) \cdot \hat{\mathbf{v}}]^2. \quad (17)$$

with $\Delta \mathbf{r}_{\mathcal{S}} = \mathbf{r}_{\mathcal{S}} - \mathbf{r}_0$ the distance from the center of the circular surface. The vector product between square brackets is recast into

$$[\hat{\mathbf{n}}_h (\Delta \mathbf{r}_{\mathcal{S}} \cdot \hat{\boldsymbol{\chi}}) \cdot \hat{\mathbf{v}}]^2 = (u_x \cos qtR + u_y \sin qtR)^2$$
$$= \sin^2 \theta \cos^2(\varphi - qtR) \quad (18)$$

where $R = e_{1z} \sin \xi + e_{2z} \cos \xi = |\sin \theta| \sin \xi$. With these simplifications, the surface anchoring free energy can be written as an explicit double integral

$$F_s = -\frac{1}{2} W_0 \sin^2 \theta \int_0^{2\pi} d\xi \int_0^{\frac{D_c}{2}} dtt \cos^2(\varphi - qt |\sin \theta| \sin \xi), \quad (19)$$

which can be solved in closed form and leads to the following expression for the surface anchoring free energy per disk

$$F_s = -\frac{\pi}{4} W_0 D_c^2 \sin^2 \theta \left( \frac{1}{2} + \cos(2\delta) \frac{J_1(qD_c |\sin \theta|)}{qD_c |\sin \theta|} \right), \quad (20)$$

with $J_1(x)$ a Bessel function of the first kind, $\delta = \varphi - qz$ the azimuthal angle with respect to the local director $\hat{\mathbf{n}}$ along the cholesteric helix (see Fig. 1). The surface anchoring amplitude per disk can be expressed in dimensionless form via $\bar{W} = \beta W_0 D_c^2$ with $\beta^{-1} = k_B T$ the thermal energy. Taking disks with diameter $D_c \approx 2\,\mu\text{m}$ and $W_0 \approx 10^{-6}-10^{-5}\,\text{Jm}^{-2}$ we find $\bar{W} \approx 10^3 - 10^4$, indicating that surface anchoring realignment is robust against thermal fluctuations in the experimental regime. We infer from Eq. (20) that the surface anchoring energy reaches a minimum at an equilibrium angle $\theta^* = \pi/2$ and $\delta^* = 0$, demonstrating preferential alignment of the disk normal along the local LC host director $\hat{\mathbf{n}}$, in agreement with experimental observation (Fig. 5). We conclude that the realignment of colloidal disks immersed in a low-molecular-weight cholesteric LC host is driven primarily by surface-anchoring forces with bulk elastic distortions around colloids playing a secondary role.

**Surface anchoring free energy of a cylindrical rod immersed in a cholesteric host.** Let us repeat the previous analysis to describe the case of a thin colloidal rod with $L_c/D_c \to \infty$ by neglecting small contributions associated with the ends of the cylinder so we only need to parameterize the cylindrical surface following the principal contour $\mathbf{r}_S = \mathbf{r}_0 + t\hat{\mathbf{u}}$ with $-L_c/2 < t < L_c/2$ of a cylinder with center-of-mass position $\mathbf{r}_0$ and long axis $\hat{\mathbf{u}}$. In this coordinate frame the surface integral in Eq. (15) is written as $\int dS = \frac{1}{2} D_c \int_0^{2\pi} d\xi \int_{-L_c/2}^{L_c/2} dt = \pi L_c D_c$. The RP surface anchoring free energy then becomes

$$F_s = -\frac{1}{4} D_c W_0 \int_0^{2\pi} d\xi \int_{-L_c/2}^{L_c/2} dt [\hat{\mathbf{n}}_h (\Delta \mathbf{r}_S \cdot \hat{\boldsymbol{\chi}}) \cdot \hat{\mathbf{v}}]^2. \quad (21)$$

The molecular LC prefers to anchor homeotropically at the rod surface so that $\hat{\mathbf{v}} = \hat{\mathbf{e}}_1 \cos\xi + \hat{\mathbf{e}}_2 \sin\xi$ is a surface normal vector in terms of the auxiliary unit vectors $\hat{\mathbf{e}}$ defined below Eq. (16). The rest of the analysis proceeds in a way analogous to the case of disks discussed previously. The dot product indicating the cosine of the angle between the surface normal and the host director locally along the rod surface parameterized by $t$ and $\xi$ is obtained from straightforward algebra

$$\hat{\mathbf{n}}_h(\Delta\mathbf{r}_S \cdot \hat{\boldsymbol{\chi}}) \cdot \hat{\mathbf{v}} = \cos(qtu_z)(e_{1x}\cos\xi + e_{2x}\sin\xi) \\ + \sin(qtu_z)(e_{1y}\cos\xi + e_{2y}\sin\xi) \quad (22)$$

The double integral over the surface contour variables $t$ and $\xi$ can be solved analytically and leads to the following expression for the surface anchoring free energy experienced by a thin colloidal rod immersed in a cholesteric molecular LC

$$F_s = -\frac{\pi}{8} L_c D_c W_0 \left( w_1 + w_2 \cos(2\delta) \frac{\sin(qL_c\cos\theta)}{qL_c} \right), \quad (23)$$

with angular coefficients $w_1 = 1 + \cos^2\theta$ and $w_2 = -\sin\theta\tan\theta$ in terms of the polar $\theta$ and azimuthal rod angle $\varphi$ with respect to the helical axis $\hat{\boldsymbol{\chi}}$.

The RP free energy is minimal at an equilibrium angle $\theta^* = 0$ (with the azimuthal angle $\varphi$ randomly distributed) which corresponds to the rod being aligned along the $\hat{\boldsymbol{\chi}}$ direction. However, there is a second, degenerate minimum at $\theta^* = \pi/2$ and $\delta^* = \pi/2$, that describes a rod pointing along the $\hat{\boldsymbol{\tau}}$-direction. The minimum surface anchoring energy is $F_s = -(\pi/4)L_c D_c W_0$ for both cases. The energy barrier between the two minima is only about 1 $k_B T$ per rod so thermal fluctuations should easily make the rods rotate within the $\hat{\boldsymbol{\chi}} - \hat{\boldsymbol{\tau}}$-plane perpendicular to the molecular director $\hat{\mathbf{n}}$ which would retain uniaxial symmetry (cf. Fig. 2c). This scenario is clearly at odds with the experimental results in Fig. 6 which indicate a strong preference for rods to align along $\hat{\boldsymbol{\tau}}$ suggesting a broken orientational symmetry within the $\hat{\boldsymbol{\chi}} - \hat{\boldsymbol{\tau}}$-plane. We thus conclude that a description based on surface-anchoring alone is inadequate and that weak elastic distortions around

the colloidal rods must be accounted for to explain the experimental situation.

**Elastic deformations around the colloidal rod surface.** So far we have completely ignored the role of weak elastic deformations of the host director ($\ell_s = K/W_0 \to \infty$) and assumed that rod realignment in a cholesteric molecular host is dominated entirely by surface anchoring effects. The experimental reality, however, is that the surface anchoring extrapolation length is large but finite ($\ell_s \approx 600\,\text{nm} \gg D_c$). Experimental observations compiled in Fig. 6 point at a scenario where rods orient preferentially along the $\tau$ direction, rather than the helical axis ($\hat{\boldsymbol{\chi}}$) as predicted from minimizing the bare RP surface anchoring energy. A possible reason as to why rod alignment along the helical axis ($\hat{\boldsymbol{\chi}}$) is strongly disfavored is that it involves a twisting of the surface disclination-like region that runs along the rod contour, which costs elastic energy. No such twisting is required if the rod points along $\hat{\boldsymbol{\tau}}$. In principle, weak director distortions may also lead to a mild decrease in the bulk nematic order parameter, particularly in regions where the director curvature is strong. In our analysis, we will assume that the bulk scalar order parameter of the host is constant throughout the system. Even in the near-field limit close to the rod surface where deviations from bulk nematic order are strongest, we expect local distortions in bulk nematic order to be minor compared to the (infinitely) strong anchoring scenario that is considered in the theoretical study by Brochard and De Gennes[60].

We now attempt to quantify the twisted disclination effect by introducing an angular deviation $\Phi(\mathbf{r})$ and express the helical host director as follows

$$\hat{\mathbf{n}}_h(\mathbf{r}) = \hat{\mathbf{x}} \cos[qz + \Phi(\mathbf{r}_\perp)] + \hat{\mathbf{y}} \sin[qz + \Phi(\mathbf{r}_\perp)], \quad (24)$$

with $\mathbf{r}$ denoting a 3D distance vector and $\mathbf{r}_\perp$ a lateral (2D) displacement vector away from the rod core perpendicular to the helical axis $\hat{\boldsymbol{\chi}}$. The total free energy of a colloidal rod inclusion aligned along the helical axis is given by the RP surface anchoring term Eq. (15) combined with the Frank elastic free energy in the presence of chirality[61]

$$F = \frac{1}{2} \int d\mathbf{r} \left[ K_{11}(\nabla \cdot \hat{\mathbf{n}}_h)^2 + K_{22}(\hat{\mathbf{n}}_h \cdot \nabla \times \hat{\mathbf{n}}_h + q)^2 \right. \\ \left. + K_{33}(\hat{\mathbf{n}}_h \times \nabla \times \hat{\mathbf{n}}_h)^2 \right] - \frac{1}{2} W_0 \oint dS(\hat{\mathbf{n}}_h \cdot \hat{\mathbf{v}}(S))^2, \quad (25)$$

with $K_{11}$, $K_{22}$ and $K_{33}$ respectively denoting the splay, twist and bend elastic modulus, as defined in our simulation model, discussed previously. For simplicity, we assume the rod to be infinitely thin and ignore distortions near the rod tips. Let us focus first on the elastic part of this free energy. Next we employ circular coordinates $\Phi(\mathbf{r}_\perp) = \Phi(r, \vartheta)$ with $r$ the lateral distance away from the rod core and $\vartheta$ an azimuthal angle so that $d\mathbf{r} = d_2\mathbf{r}_\perp dz$ and $d_2\mathbf{r}_\perp = r dr d\vartheta$ and expand up to second order in $q$. Since the distortion pattern does not change along the rod direction the integration over $z$ is trivial and we obtain for the elastic part of the free energy $F_{el}$ per unit rod length

$$\frac{F_{el}}{L_c} = \frac{1}{2} \int d_2\mathbf{r}_\perp \left\{ \frac{K_{11}}{r^2}(1 + \partial_\vartheta\Phi)^2 + K_{33}(\partial_r\Phi)^2 \right. \\ \left. + \frac{(qL_c)^2}{12} \Delta K \left[ \frac{1}{r^2}(1 + \partial_\vartheta\Phi)^2 - (\partial_r\Phi)^2 \right] \right\}, \quad (26)$$

where $\Delta K = K_{33} - K_{11} > 0$ denotes the difference between the bend and splay moduli. The elastic anisotropy turns out to be of crucial importance since the twist correction $\mathcal{O}(q^2)$ vanishes in case of the one-constant approximation $K_{11} = K_{33} = K_{22} = K$.

The change of elastic free energy induced specifically by the helical director twist is given by the second term in the above

expression

$$\Delta F_{\text{twist}}^{(el)} = \frac{1}{24}(qL_c)^2 L_c \Delta K \mathcal{F}[\Phi_0], \tag{27}$$

where $\Phi_0$ denotes the distortion angle for the untwisted system, and

$$\mathcal{F}[\Phi_0] = \int d_2 \mathbf{r}_\perp \left[ \frac{1}{r^2}(1 + \partial_\vartheta \Phi_0)^2 - (\partial_r \Phi_0)^2 \right], \tag{28}$$

is a dimensionless quantity measuring the extent of the surface disclination surrounding the cylinder. Applying the one-constant approximation which does not lead to qualitative changes in this context, we determine $\Phi_0$ from minimizing

$$\frac{F_{el}(q=0)}{KL_c} = \frac{1}{2} \int d_2 \mathbf{r}_\perp \left\{ \frac{1}{r^2}(1 + \partial_\vartheta \Phi)^2 + (\partial_r \Phi)^2 \right\}, \tag{29}$$

so that $(\delta F_{el}/\delta \Phi)_{\Phi_0} = 0$ and $\ell_s = K/W_0$ defines the surface anchoring extrapolation length. Functional minimization of the free energy we obtain the Laplace equation in polar coordinates

$$\partial_r^2 \Phi_0 + \frac{1}{r}\partial_r \Phi_0 + \frac{1}{r^2}\partial_\vartheta^2 \Phi_0 = 0, \tag{30}$$

Let us now turn to the surface anchoring energy given by the last contribution in Eq. (25). Ignoring the effect of chirality on the surface anchoring energy (Supplementary) and parameterizing the surface normal vector $\hat{\mathbf{v}}(\mathcal{S}) = \cos\vartheta\,\hat{\mathbf{x}} + \sin\vartheta\,\hat{\mathbf{y}}$ in terms of the azimuthal angle $\vartheta$ we find

$$F_s = -\frac{1}{2}W_0 L_c \oint_{\mathcal{C}} d\vartheta \cos^2(\vartheta - \Phi_0), \tag{31}$$

The surface integral is now reduced to a line integral along the circular rod contour $\mathcal{C}$ which ensures that the interior of the rod cross-section is excluded from the spatial integration.

The above expression for the surface anchoring energy translates into the following condition for the distortion angle $\Phi_0$ at the rod surface

$$K\partial_r \Phi_0(D_c/2, \vartheta) - \frac{1}{4}W_0 \sin\left[2(\vartheta - \Phi_0(D_c/2, \vartheta))\right] = 0 \tag{32}$$

In the limit of weak surface anchoring $\ell_s \gg D_c$ relevant to our case we approximate $\sin 2(\vartheta - \Phi_0) \approx \sin 2\vartheta$ and require the distortion angle to vanish far away from the rod surface. The boundary conditions for the Laplace equation Eq. (30) are then given by

$$\begin{aligned} \Phi_0(\infty, \vartheta) &= 0 \\ \partial_r \Phi_0(D_c/2, \vartheta) &= (4\ell_s)^{-1}\sin 2\vartheta, \end{aligned} \tag{33}$$

with the latter denoting a Neumann boundary condition at the surface of the circular rod cross section. The result is a simple dipolar field

$$\Phi_0(r, \vartheta) = -\frac{D_c}{16\ell_s}\left(\frac{D_c}{2r}\right)^2 \sin 2\vartheta. \tag{34}$$

The distortion pattern associated with Eq. (34) is in good agreement with the one obtained from simulations (Fig. 3b).

Plugging the dipolar form back into Eq. (28) and integrating we find that the difference in elastic energy between the twisted ($\hat{\chi}$) and untwisted ($\hat{\tau}$) alignment directions in independent of the surface anchoring extrapolation length $\ell_s$ but depends sensitively on the colloidal rod length and elastic anisotropy of the twisted host

$$\Delta F_{\text{twist}}^{(el)} \sim \frac{\pi}{12}(qL_c)^2 L_c \Delta K \ln\left(\frac{2\ell_{\text{cut}}}{D_c}\right). \tag{35}$$

Setting $\ell_{\text{cut}} = D_c$ as a typical length-scale for the extent of the elastic distortion surrounding the rod core, and using a splay-bend elastic anisotropy $\Delta K = 4\text{pN}$ we find that $\Delta F_{\text{twist}} \approx 38k_{\text{B}}T$ suggesting that the energy associated with the elastic distortion is far greater than the thermal energy and that the twisted disclinations are strong enough to break the degeneracy of orientations in the $\hat{\tau} - \hat{\chi}$-plane, as reflected in Fig. 2c and Fig. 6c. As a more general consideration, we may equate the energy scale imparted by Eq. (35) with the thermal energy $\Delta F_{\text{twist}}^{(el)} = k_{\text{B}}T$ to obtain the criterion Eq. (1) for the minimum chiral strength that is required for the twisted disclination effect to have an impact on the phase behavior and to access the supercritical region of the phase diagram for chiral hybrid LCs shown in Fig. 2f. The current model only provides an approximate estimate for the energy scale associated with elastic distortion incurred by the chiral host as it is based on the assumption that the local host nematic order parameter $S_m$ is constrained at its far-field bulk value and is not allowed to relax in regions where director distortions are the largest, as observed in experiment and simulation.

In principle, Eq. (35) needs to be complemented with contributions from the surface anchoring energy (last term in Eq. (25)) as well as from surface elasticity (saddle-splay, not included in Eq. (25)). Both effects are found to be very weak indeed as discussed in Supplementary and shall be further ignored in our discussion. In Supplementary we address weak elastic distortions that are incurred when the rod remains perpendicular to the helical axis $\hat{\chi}$ but is allowed to display angular fluctuations in the $\hat{\mathbf{n}} - \hat{\tau}$-plane. The results are compiled in Fig. 7. With numerical and theoretical methods found in good agreement, we conclude that the energy associated with the elastic distortions is strongly inferior to the ones generated by surface-anchoring effects. Except for very weak surface anchoring strengths the energy penalty preventing rods from co-aligning with the molecular director $\hat{\mathbf{n}}$ widely exceeds the thermal energy. Gathering the findings of the previous paragraphs, we summarize the realigning potential acting on a rod immersed in a cholesteric host in Results Eq. (4).

## Data availability

All data supporting the findings of this study are available within the article and its Supplementary files. The unprocessed images of this study are available without restrictions and can be obtained by request to the corresponding author via email. Source data are provided with this paper.

## Code availability

The code for the molecular LC simulation can be found in our previous work[17].

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

## Acknowledgements

We acknowledge discussions with M. Bowick, T. Lee, S. Ghosh, L. Longa, T.C. Lubensky, B. Senyuk, M. Ravnik, and M. Tasinkevych. We are grateful to L. Longa and T.C. Lubensky for providing helpful suggestions and feedback on the initial versions of this manuscript. The experimental and numerical simulations research at the University of Colorado Boulder was supported by the US Department of Energy, Office of Basic Energy Sciences, Division of Materials Sciences and Engineering, under contract DE-SC0019293 with the University of Colorado at Boulder. M.T.L. and H.H.W. acknowledge financial support from the French National Research Agency (ANR) under grant ANR-19-CE30-0024 "ViroLego". I.I.S. acknowledges the support of the International Institute for Sustainability with Knotted Chiral Meta Matter at Hiroshima University in Japan during part of his sabbatical stay, as well as the hospitality of the Kavli Institute for Theoretical Physics in Santa Barbara, when he was partially working on this manuscript while supported in part by the National Science Foundation under Grant No. NSF PHY-1748958 (I.I.S. and J.-S.W.).

## Author contributions

H.M. performed experiments. J.-S.W. analyzed experimental data and performed numerical modeling. M.T.L. and H.H.W. performed analytical modeling. J.-S.W., H.H.W., and I.I.S. wrote the manuscript, with input and feedback from all authors. I.I.S. designed and directed the project.

## Competing interests

The authors declare no competing interests.
