## [Transparent Peer Review file · Nature Communications]

Emergent biaxiality in chiral hybrid liquid crystals

Corresponding Author: Professor Ivan Smalyukh

Version 0:

Reviewer comments:

Reviewer #1

(Remarks to the Author)

The present work is mostly an experimental study in which biaxiality in colloidal liquid crystals is studied.

The most noteworthy result obtained is the unambiguous demonstration of strong biaxiality in a system of colloidal rods with special surface anchoring doped into a chiral host phase. Even for a non-experimentalist like myself the significance of this study is quite clear. I consider this paper a highly significant contribution to the field of biaxial nematic liquid crystals on account of the experiments that have been carried out. I could not detect any obvious flaws in the arguments and interpretation of the data presented.

Unfortunately, I have to say that this applies only to the experimental data but not to the parallel theoretical approaches. Here most of the derivations are unclear, largely due to the fact that symbols are generally not introduced. Because no intermediate steps in those derivations are given or even justified it is next to impossible. This negative comment applies mostly to the supplementary material presented in Sec. F where, for example, the development between Eqs. (11) - (14) remains mysterious. You should give a lot more information in Sec. F about what is the logic here and substantially more mathematical details. Otherwise, it is impossible to extract any useful information from the theoretical developments. On the contrary, I am afraid that if this section is written properly it might very well blow the paper out of proportion. As a solution to this dilemma the authors might consider putting the theoretical considerations into a separate publication.

Another problematic issue from my point of view is the use of the term „computer simulation“. As the authors are not giving any information concerning the specific simulation technique they are using (at least I couldn't find that information neither in the main body of the text nor in the supplementary material) one has no chance to validate the claims made by the authors to what an extent the theory does, in fact, support the experimental findings. In turn, it is completely impossible to verify any of the theoretical part independently.

In summary, this is quite an interesting paper that deserves publication at some point with the proviso that my above very critical comments on the theory will be met successfully in a forthcoming revised paper. As it stands at the time of writing I find the present version of the manuscript unacceptable for publication.

Reviewer #2

(Remarks to the Author)

I have examined the paper by JS Wu et al. entitled « Emergent biaxiality in chiral hybrid liquid crystals» submitted in Nature Communications.

In this work, the authors examine the biaxiality of hybrid liquid crystals made of uniaxial solid particles homogeneously dispersed in a chiral nematic liquid crystal. They found that the inclusions give to the composite a strong biaxiality even at low concentrations ($> \sim 0.2\%$). Using a combination of experiments numerical simulations and analytical approaches, they characterize the hybrids and explain their main features. The paper is quite well written and richly illustrated. The data are obtained by very convincing methods and are discussed in depth. The conclusions are clear and sound correct.

In the last 10 years the group of Smalyukh has developed many original and innovative systems made of home-made colloids dispersed in liquid crystals. Contrary to composite systems described by many groups, these hybrids form stable mesophases and can be studied as model systems. They show very exotic structures and properties. Here the authors focus on the interplay between the chirality of the matrix, the alignment distribution of the inclusions and the biaxial structure of the resulting phase. If the interplay between chirality and biaxiality has been examined in purely molecular mixtures or colloidal

mixtures, the effect was weak. It is only because of the large scale difference between molecules and inclusions (but both in thermal equilibrium) that a huge effect is observed in these new types of hybrid systems. I am not aware of a similar study and I think it could be interesting for the condensed matter community (not only the liquid crystal one). Due to the quality of the research and the originality of the effect, I recommend publication in Nature Communications.

I have only minor remarks and noted a few typos.

- Fig2 is early introduced to give evidence of the properties of the "Supercritical biaxial colloidal fluid." Many parameters are thus undefined yet. If most of them can be understood easily, the definition of the order parameters is not given. I only found "The details of the model are given in Supplementary" in the main text when calling Fig. 2(e,f). I think that the authors should also send the reader to Section VII for the definitions of the two order parameters (either in the caption or in the main text).

- "demonstrating different energy barriers for the thin disk fluctuating in different angular directions." Maybe "energy wells" would be better, since the fluctuations probe the shape of the potential energy well (rather than the barrier around)?

- (If I am correct) I found that the fluctuation angles δ ζ are not coherently defined between the graphs and the double arrows given in the illustration sketches. For example, the double arrow related to δ in Fig3C does not start from the director direction but rather seem to reflect the angular distribution (I ask this also because the graphs stop are shown in the range $[-45^\circ, 45^\circ]$). Please clarify

- P7: "The total external potential is given by the bare Rapini-Papoular (RP) contribution for the undistorted host director plus the free energy contributions from elastic distortions". Maybe replace by ~ "The total external potential is given by the bare Rapini-Papoular (RP) surface anchoring contribution for the undistorted host director plus the free energy contributions from bulk elastic distortions" in order to give a glimpse to non-experts.

- P 17 "and r_\perp the lateral distance perpendicular to the helical axis" (lateral distance vector ?)

- P 18 "The distortion pattern associated with Eq. (25) is in good agreement with the one obtained from simulations [ExtFig. 1(b)]." The authors could clarify this passage since Ext Fig.1b corresponds to a numerical simulation for a rod aligned along τ whereas the analytical approach starts with a rod aligned along the helix (the passage might be confusing even if I understand that the solution ϕ_0 corresponding to the untwisted system is closed to the one for the rod aligned along τ with $z \leftrightarrow y$).

- P18: "the energy barrier preventing rods from co-aligning with the molecular director \hat{n} widely exceeds the thermal energy. " Here also (but I might be wrong if I missed something) ExtFig3 examines the energy well around the energy minimum and does not show any energy barrier, only a monotonic increase in the considered range. I expect that the rod aligned along n is at a maximum (at least for surface energy) and not another local minimum (as in Eq.4). So I would not use energy barrier here for clarity.

- P22 caption of Fig Ext 2: units are absent with the value of anchoring strength (also a typo "align long")

Version 1:

Reviewer comments:

Reviewer #1

(Remarks to the Author)

I would like to thank the authors for picking up my earlier criticism. I have been reading the revised version of the paper quite thoroughly and must say that I am quite satisfied with the result of the revision. In fact, the paper has improved so much from its original version that one may ask: why didn't you do it like this in the first place? Anyway, this is not an argument for a scientific discussion and because I do no longer uphold the editorial process any further of this work I now recommend publication of the paper in its present form.

Reviewer #2

(Remarks to the Author)

I think that the authors have taken into account all the comments i have made in the previous round. I still think that this work deserves publication due the quality of the research and the originality of the effect described here, that could be interesting for a broad condensed matter audience.

Manuscript NCOMMS-24-35270-T
Emergent biaxiality in chiral hybrid liquid crystals
Response to reviewer questions and summary of changes

J.-S. Wu et al.

We thank both referees for their efforts on reviewing our work and for their constructive comments. Below we provide a point-by-point response to their queries which have helped us to clarify parts of our analytical and simulation models, correct some omissions and improve the overall consistency of our presentation. All changes are marked in red in our manuscript.

Reply to REVIEWER 1

Reviewer: *The present work is mostly an experimental study in which biaxiality in colloidal liquid crystals is studied. The most noteworthy result obtained is the unambiguous demonstration of strong biaxiality in a system of colloidal rods with special surface anchoring doped into a chiral host phase. Even for a non-experimentalist like myself the significance of this study is quite clear. I consider this paper a highly significant contribution to the field of biaxial nematic liquid crystals on account of the experiments that have been carried out. I could not detect any obvious flaws in the arguments and interpretation of the data presented.*

Authors: We thank the referee for a careful review of our manuscript and finding our work unambiguous, clear, and a highly significant contribution.

Reviewer: *Unfortunately, I have to say that this applies only to the experimental data but not to the parallel theoretical approaches. Here most of the derivations are unclear, largely due to the fact that symbols are generally not introduced. Because no intermediate steps in those derivations are given or even justified it is next to impossible. This negative comment applies mostly to the supplementary material presented in Sec. F where, for example, the development between Eqs. (11) - (14) remains mysterious. You should give a lot more information in Sec. F about what is the logic here and substantially more mathematical details. Otherwise, it is impossible to extract any useful information from the theoretical developments. On the contrary, I am afraid that if this section is written properly it might very well blow the paper out of proportion. As a solution to this dilemma the authors might consider putting the theoretical considerations into a separate publication.*

Authors: We agree the derivation of the surface energies in Methods F was insufficiently clear, not least because of some errors and omissions. These mostly concerned the considerations leading up to the disk surface anchoring free energy Eq. (15) from its formal (RP) definition Eq. (11) (equation numbers refer to the original manuscript) that the reviewer correctly pointed out as being “mysterious”. We have revised the discussion and provided additional mathematical detail to clarify the technical parts of the derivation of the surface anchoring energy Eq. (20) (revised manuscript). The same goes for the surface anchoring energy of a colloidal rod in a molecular cholesteric (discussion leading up to Eq. (23)) where the analysis is largely analogous to the case of the disks presented in the preceding paragraph. A thorough revision of Supplementary C (Supplementary III in the revised version) has been carried in order to improve clarity of the argumentation and notation used there. Furthermore, we have provided more context in the discussion around Eqs. (26) and (33) as well as added an important cross-reference between after Eq. (35) relating to Eq. (1) in the main text. Throughout our discussion we have thoroughly verified all expressions to assure all symbols were properly introduced and we have rectified matters where necessary.

We do wish to underline that the main purpose of the analytical theory presented in Methods is to point out the main steps regarding the physics and basic geometry of the problem as well as to substantiate the various approximations to keep the theory tractable and insightful. We have refrained from showing elementary mathematical details, for instance, those related to solving standard multiple integrals or Laplace equations, Taylor expansions up to leading order, and switching between Cartesian and cylindrical coordinate frames. These are all important but fairly standard operations explained in most LC and mathematical textbooks. The same holds for a background discussion on starting expressions such as the ones describing (Rapini-Papoular-type) surface anchoring and Frank elasticity.

The principal logic behind the analytical theory is to make the case that surface anchoring alone cannot account for our experimental observations of rod-based chiral hybrid LCs but that a subtle coupling with bulk elastic distortions compounded by the twisted nature LC background is an essential ingredient for understanding the orientational

order of thin colloid rods immersed in such complex environments. The case of disks is presented first and foremost as a benchmark counterexample where the Rapin-Papoular surface anchoring description [Eq. 20] *does* suffice to account for the experimental observations which suggests that bulk elastic distortions are immaterial for explaining single-disk orientational order.

Reviewer: *Another problematic issue from my point of view is the use of the term “computer simulation”. As the authors are not giving any information concerning the specific simulation technique they are using (at least I couldn’t find that information neither in the main body of the text nor in the supplementary material) one has no chance to validate the claims made by the authors to what an extent the theory does, in fact, support the experimental findings. In turn, it is completely impossible to verify any of the theoretical part independently.*

Authors: We appreciate this remark. All essential aspects and details of the numerical simulation for LC host medium are given in the new Methods section E, into which we transferred an expanded description that we had in the Supplementary E in our previous version. We have also added a reference to our previous work (with a Matlab script provided in the supplementary of this work), as well as a reference to a classical detailed review on these types of simulations (ref. 53) and several other works where such modeling was performed, so that readers can have a self-contained description within the current manuscript and additional references to follow up with in case of the interest to learn even more about the modeling. We hope all the information regarding computer simulation is now clear and easy to find.

Reviewer: *In summary, this is quite an interesting paper that deserves publication at some point with the proviso that my above very critical comments on the theory will be met successfully in a forthcoming revised paper. As it stands at the time of writing I find the present version of the manuscript unacceptable for publication.*

Authors: We thank the referee for the positive remarks on the content of our work as well as for pointing out some deficiencies in the modelling sections. We hope we were able to successfully address these remarks and suggestions.

Reply to REVIEWER 2

Reviewer: *I have examined the paper by JS Wu et al. entitled “Emergent biaxiality in chiral hybrid liquid crystals” submitted in Nature Communications. In this work, the authors examine the biaxiality of hybrid liquid crystals made of uniaxial solid particles homogeneously dispersed in a chiral nematic liquid crystal. They found that the inclusions give to the composite a strong biaxiality even at low concentrations ($\geq 0.2\%$). Using a combination of experiments numerical simulations and analytical approaches, they characterize the hybrids and explain their main features. The paper is quite well written and richly illustrated. The data are obtained by very convincing methods and are discussed in depth. The conclusions are clear and sound correct.*

In the last 10 years the group of Smalyukh has developed many original and innovative systems made of home-made colloids dispersed in liquid crystals. Contrary to composite systems described by many groups, these hybrids form stable mesophases and can be studied as model systems. They show very exotic structures and properties. Here the authors focus on the interplay between the chirality of the matrix, the alignment distribution of the inclusions and the biaxial structure of the resulting phase. If the interplay between chirality and biaxiality has been examined in purely molecular mixtures or colloidal mixtures, the effect was weak. It is only because of the large scale difference between molecules and inclusions (but both in thermal equilibrium) that a huge effect is observed in these new types of hybrid systems. I am not aware of a similar study and I think it could be interesting for the condensed matter community (not only the liquid crystal one). Due to the quality of the research and the originality of the effect, I recommend publication in Nature Communications.

Authors: We thank the reviewer for providing a detailed report on our work. We are pleased to hear that our presentation is “well written and richly illustrated” and that the findings compiled in our manuscript are of quality and originality.

Reviewer: *I have only minor remarks and noted a few typos.*

- Fig2 is early introduced to give evidence of the properties of the “Supercritical biaxial colloidal fluid.” Many parameters are thus undefined yet. If most of them can be understood easily, the definition of the order parameters is not given. I only found “The details of the model are given in Supplementary” in the main text when calling

Fig. 2(e,f). I think that the authors should also send the reader to Section VII for the definitions of the two order parameters (either in the caption or in the main text).

Authors: We appreciate these helpful suggestions, which we fully accounted for. We have defined the colloidal order parameters and the chiral wavenumber in the caption and provided a reference to Section VII for a detailed discussion on the colloidal order parameters.

Reviewer:- *“demonstrating different energy barriers for the thin disk fluctuating in different angular directions.” Maybe “energy wells” would be better, since the fluctuations probe the shape of the potential energy well (rather than the barrier around)?*

Authors: We agree this is a more appropriate term that we have now adopted it.

Reviewer:- *(If I am correct) I found that the fluctuation angles δ ζ are not coherently defined between the graphs and the double arrows given in the illustration sketches. For example, the double arrow related to δ in Fig3C does not start from the director direction but rather seem to reflect the angular distribution (I ask this also because the graphs stop are shown in the range $[-45^\circ, 45^\circ]$). Please clarify*

Authors: We appreciate this remark. We have clarified that we define the colloidal particle angles (δ and ζ for disk; η and γ for rod) such that they all start from the average colloidal director orientation and span equally to + and - sides. Here for disks, we use the disk normal direction (perpendicular to disk surface) as colloidal director. (here it also coincides with the molecular director $\hat{\mathbf{n}}$). By such means the distribution peak would be at the center of graph and we are able to crop out insignificant tails by displaying a smaller range, such as $[-45^\circ, 45^\circ]$.

Reviewer:- *P7: “The total external potential is given by the bare Rapini-Papoular (RP) contribution for the undistorted host director plus the free energy contributions from elastic distortions”. Maybe replace by “The total external potential is given by the bare Rapini-Papoular (RP) surface anchoring contribution for the undistorted host director plus the free energy contributions from bulk elastic distortions” in order to give a glimpse to non-experts.*

Authors: We appreciate this suggestion and have corrected this sentence accordingly.

Reviewer:- *P 17 “and r_\perp the lateral distance perpendicular to the helical axis” (lateral distance vector ?)*

Authors: We have corrected this.

Reviewer:- *P 18 “The distortion pattern associated with Eq. (25) is in good agreement with the one obtained from simulations [ExtFig. 1(b)].” The authors could clarify this passage since Ext Fig.1b corresponds to a numerical simulation for a rod aligned along τ whereas the analytical approach starts with a rod aligned along the helix (the passage might be confusing even if I understand that the solution ϕ_0 corresponding to the untwisted system is closed to the one for the rod aligned along tau with $z \leftrightarrow y$).*

Authors: We appreciate this remark. This is a correct observation. Indeed, the lateral distortions incurred by the presence of a thin rod as depicted in the cross section patterns in ExtFig. 1b (Fig. 3 in the revised manuscript) are the same whether the rod points along $\hat{\tau}$ or $\hat{\chi}$. The subtle difference is that this pattern is weakly twisted along the rod contour in one case ($\hat{\chi}$) but not the other ($\hat{\tau}$). Since the chiral twist is very weak on the scale of the rod diameter $qD_c \approx 0.006$ these patterns are not affected by the twist. However, given that rods are very long compared to their diameter the energy penalty incurred by twisting the distortion pattern along the rod contour strongly exceeds the thermal energy (discussed in Methods). We have carefully updated text and assured that this is clear to readers.

Reviewer:- *P18: “the energy barrier preventing rods from co-aligning with the molecular director n widely exceeds the thermal energy. “ Here also (but I might be wrong if I missed something) ExtFig3 examines the energy well around the energy minimum and does not show any energy barrier, only a monotonic increase in the considered range. I expect that the rod aligned along n is at a maximum (at least for surface energy) and not another local minimum (as in Eq.4). So I would not use energy barrier here for clarity.*

Authors: We appreciate this remark. We have replaced it more appropriately by “energy penalty”.

Reviewer:- *P22 caption of Fig Ext 2: units are absent with the value of anchoring strength (also a typo "align long")*

Authors: We have fixed these typos and omissions.

Authors: We thank the reviewer once again for a detailed examination of our manuscript as well as for providing helpful suggestions for improvement that we have all followed up on.